# TINKER: DIFFUSION'S GIFT TO 3D—MULTI-VIEW CONSISTENT EDITING FROM SPARSE INPUTS WITHOUT PER-SCENE OPTIMIZATION

**Canyu Zhao**[1*]  **Xiaoman Li**[1*]  **Tianjian Feng**[1]  **Zhiyue Zhao**[1]  **Hao Chen**[1]  **Chunhua Shen**[1,2,3]

[1] Zhejiang University, State Key Lab of CAD & Computer Graphics, China  [2] Ant Group
[3] Zhejiang University of Technology, China

## ABSTRACT

We introduce TINKER, a novel framework for high-fidelity 3D editing without any per-scene finetuning, where only a single edited image (one-shot) or a few edited images (few-shot) are required as input. Unlike prior techniques that demand extensive per-scene optimization to ensure multi-view consistency or to produce dozens of consistent edited input views, TINKER delivers robust, multi-view consistent edits from as few as one or two images. This capability stems from repurposing pretrained diffusion models, which unlocks their latent 3D awareness. To drive research in this space, we curate the first large-scale multi-view editing dataset and data pipeline, spanning diverse scenes and styles. Building on this dataset, we develop our framework capable of generating multi-view consistent edited views without per-scene training, which consists of two novel components: (1) Multi-view consistent editor: Enables precise, reference-driven edits that remain coherent across all viewpoints. (2) Any-view-to-video scene completion model : Leverages spatial-temporal priors from video diffusion to perform high-quality scene completion and novel-view generation even from sparse inputs. Through extensive experiments, TINKER significantly reduces the barrier to generalizable 3D content creation, achieving state-of-the-art performance on editing, novel-view synthesis, and rendering enhancement tasks, while also demonstrating strong potential for 4D editing. We believe that TINKER represents a key step towards truly scalable, zero-shot 3D and 4D editing.

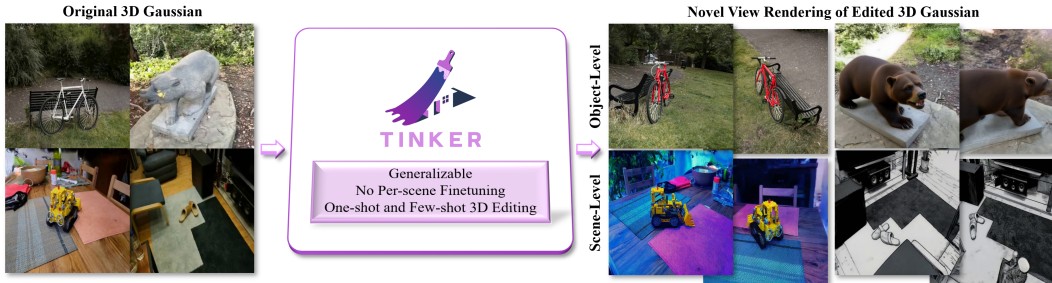

Figure 1: Compared with prior 3D editing approaches, TINKER removes the necessity of labor-intensive per-scene fine-tuning——whether for generating multi-view-consistent edited inputs for 3DGS optimization or for preserving consistency through scene-specific training. Moreover, TINKER is capable of performing both object-level and scene-level 3D editing, and achieves high-quality results in few-shot as well as one-shot settings. **Please refer to Figures S2 for 4D editing results and Figures S8, S9, S10, S11 for more compelling visualizations.**

---

*CZ and XL claim equal contributions. CS is the corresponding author.

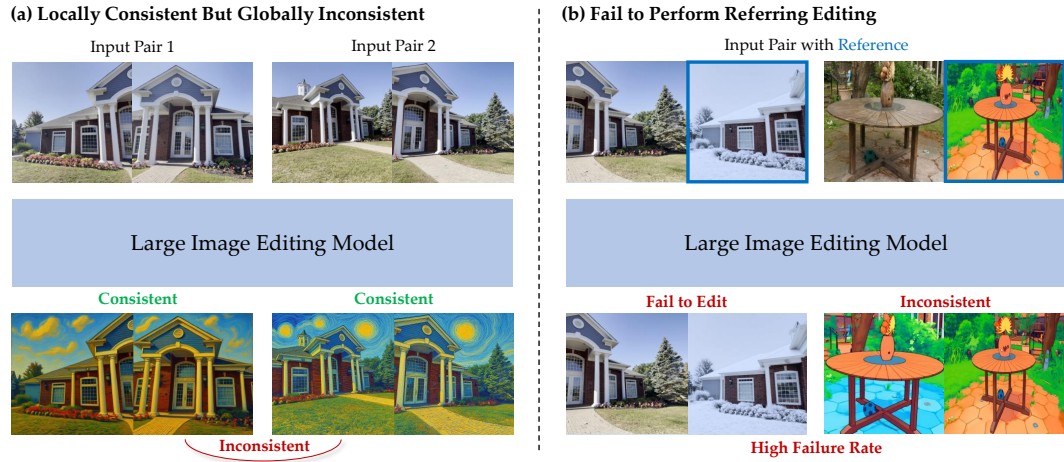

Figure 2: (a) Recent large image editing model achieves multi-view consistent image editing by horizontally concatenating two images and editing them jointly. Although it ensures consistency between the concatenated image pair, significant inconsistencies remain across different image pairs. (b) We also evaluated whether recent large image editing model can edit one half of the concatenated image by referencing the other half. The results demonstrate that the model lacks this capability.

# 1 INTRODUCTION

Benefiting from the rapid advancements in 2D diffusion models (Rombach et al., 2022; Esser et al., 2024; Labs, 2024), a prevailing paradigm for 3D editing has emerged: generating multi-view consistent images via 2D diffusion-based editing, followed by fine-tuning 3D Gaussian Splatting (3DGS) (Kerbl et al., 2023) or Neural Radiance Field (NeRF) (Mildenhall et al., 2021) to edit 3D scenes. This pipeline has become the de facto standard in recent 3D editing approaches.

During the era when U-Net-based diffusion models dominated, many successful 3D editing approaches were inspired by advances in 2D image editing, such as Instruct-NeRF2NeRF (Haque et al., 2023) and InstructPix2Pix (Brooks et al., 2023). Recently, the emergence of Diffusion Transformer (DiT) architectures (Peebles & Xie, 2023) and Flow Matching (Liu et al., 2022; Lipman et al., 2022; Albergo & Vanden-Eijnden, 2022; Esser et al., 2024) has significantly advanced the field of generative modeling. Latest developments have demonstrated substantial improvements in both image and video generation (Wan et al., 2025; Esser et al., 2024; Labs, 2024), editing (Wang et al., 2024a; Yu et al., 2025; Ku et al., 2024; Jiang et al., 2025; Labs et al., 2025), and even vision understanding (Zhao et al., 2025; Wang et al., 2025b; Ke et al., 2024), all driven by large-scale DiT flow-based models. Theoretically, 3D editing should also naturally evolve to incorporate and benefit from these powerful new architectures and methodologies. However, we find that current 3D editing methods have yet to fully capitalize on these recent breakthroughs. Many recent approaches, despite producing impressive results, remain constrained by conventional U-Net-based methodology (Haque et al., 2023; Wu et al., 2024b; Zhuang et al., 2024; Fujiwara et al., 2024), rather than embracing the more powerful and scalable techniques (Wang et al., 2025a; Zhang et al., 2025). Consequently, these methods require laborious per-scene fine-tuning, including model training or costly scene-by-scene hyperparameter tuning, and lack the ability to generate edits with better diversity and visual quality now expected from state-of-the-art 2D diffusion models. There remains a noticeable disconnect between the progress made in 2D editing and the latest methods in 3D editing. One key reason behind this limitation is the lack of multi-view consistent image editing datasets. For recent unified generation and editing models, fine-tuning with large-scale data has proven to be highly effective. However, the difficulty in collecting high-quality multi-view consistent datasets hinders progress in 3D-aware or view-consistent editing tasks.

Inspired by the remarkable capabilities of large language models in addressing unseen tasks (Wei et al., 2021; Achiam et al., 2023; Guo et al., 2025), we pose a natural question: can recent large-scale image editing foundation model (Labs et al., 2025) also perform multi-view consistent editing that benefits 3D editing? The answer is confirmative. We observe that simply concatenating two images as input enables these models to produce highly consistent and high-quality edits across views.

However, while this pairwise concatenation ensures consistency between the two input views, we find that significant discrepancies often arise between different image pairs, thereby limiting global view consistency. A straightforward idea is to concatenate an unedited image with an edited one, using the latter as a reference to guide the editing process. However, we observe that current foundation model does not exhibit this capability, usually producing inconsistent results, and typically reproduces the unedited image without modification. We illustrate both cases in Figure 2.

To address this issue, we design a novel pipeline that amplifies the model's capability for multi-view consistent editing. Specifically, we first introduce a data pipeline to generate referring editing dataset, where an unedited image is concatenated with another view that has already been edited. Fine-tuning with this dataset encourages the image editing model to learn how to propagate the editing intent across different viewpoints, which significantly improves the editing success rate and promotes better cross-view consistency. Furthermore, to efficiently propagate edits from a sparse set of edited views to a dense set of novel views, we introduce a scene completion model, which effectively bridges 2D and 3D editing by leveraging video editing. The model can generate a multitude of view-consistent edits from sparse inputs, thereby providing a robust data foundation for data-intensive 3DGS and ensuring high-quality results. Unlike previous approaches that rely on repeated fine-tuning either to enforce multi-view consistency or to obtain multi-view-consistent input views for downstream 3DGS optimization, **the key distinction of TINKER lies in its ability to directly produce high-quality, multi-view-consistent edited input views that can be seamlessly leveraged for 3DGS optimization without per-scene fine-tuning.** Additionally, by fully exploiting the priors embedded in foundation models, our approach is also able to enhance the overall rendering quality of 3D scenes. Furthermore, we demonstrate the significant potential of TINKER for applications in 4D editing. We believe TINKER paves the way for future research in generalizable, user-friendly 3D content creation.

In summary, our main contributions are as follows:

- We design a novel pipeline that effectively elicits the multi-view consistent editing capabilities of large-scale image editing models, and introduce, to the best of our knowledge, the first multi-view consistent image editing dataset.

- We introduce a sparse-view scene completion model specifically tailored for 3D editing tasks by rethinking editing problem as reconstruction problem. In addition to 3D editing, our model is capable of performing video reconstruction and 4D editing.

- Our TINKER eliminates the requirement for per-scene optimization that previous methods necessitate to ensure multi-view consistency or to generate multi-view-consistent edited input views, thereby significantly lowering the barrier for practical use of 3D editing. We hope that TINKER can serve as a general-purpose foundation for future advancements in 3D editing.

## 2 RELATED WORK

### 2.1 DIFFUSION MODEL

Diffusion models (Ho et al., 2020; Song et al., 2020; Rombach et al., 2022; Esser et al., 2024; Labs, 2024) are a powerful class of generative models that produce high-quality and diverse outputs by learning to reverse a gradual noising process. This process consists of a forward stage, where data is incrementally corrupted by Gaussian noise over multiple steps, and a reverse stage, where a neural network is trained to iteratively denoise and reconstruct the original data. Transformer-based architectures (Peebles & Xie, 2023; Vaswani et al., 2017) and flow-matching (Albergo & Vanden-Eijnden, 2022; Lipman et al., 2022; Liu et al., 2022) objectives have recently become the mainstream design choices in diffusion models (Esser et al., 2024; Labs, 2024), offering significant improvements in generation quality and scalability. Owing to their strong generative priors acquired by large-scale training, diffusion models have significantly advanced a variety of vision tasks, such as image and video generation (Blattmann et al., 2023; Wan et al., 2025), editing (Brooks et al., 2023; Yu et al., 2025; Labs et al., 2025; Tian et al., 2025), image perception (Ke et al., 2024; Zhao et al., 2025; Wang et al., 2025b).

## 2.2 2D EDITING

To achieve image editing, some pioneering studies explored alterations to the attention mechanism within the generative model (Hertz et al., 2022; Chefer et al., 2023). A majority of approaches in image editing primarily revolved around inversion-based methods. These methods work by first inverting an input image back into its latent noise representation, and subsequently use a new prompt to generate the edited image (Mokady et al., 2023; Cao et al., 2023; Song et al., 2020; Wang et al., 2024a; Rout et al., 2024). Beyond the inversion-based methods, some methods directly train models to follow explicit editing instructions (Brooks et al., 2023; Pan et al., 2023). Similarly, the paradigm of video editing largely aligns with that of image editing (Qin et al., 2024; Liew et al., 2023; Liu et al., 2024; Geyer et al., 2023; Ku et al., 2024; Khachatryan et al., 2023). Recently, high-quality unified models have emerged in both image (Yu et al., 2025; Labs et al., 2025) and video editing (Jiang et al., 2025). However, we observe that most mainstream approaches do not focus on multi-view consistent editing (Liu et al., 2023c;b). While some recent methods (Jiang et al., 2025) can perform depth-conditioned video editing, their primary focus lies in generation, which often results in videos with large motion and dynamics, causing multi-view inconsistencies.

## 2.3 3D EDITING

3D Gaussian Splatting (3DGS) (Kerbl et al., 2023) and Neural Radiance Fields (NeRF) (Mildenhall et al., 2021) are two widely adopted 3D representations in recent years. Early approaches typically performed style transfer by learning a mapping between the source and target scenes (Liu et al., 2023a). With the rapid advancement of 2D diffusion models, numerous 3D editing methods have incorporated them as key modules (Chen et al., 2024a; Fujiwara et al., 2024; Haque et al., 2023; Wu et al., 2024b; Zhuang et al., 2024). Some methods (Chen et al., 2024b; Decatur et al., 2024; Dong et al., 2024; Sella et al., 2023) leverage Score Distillation Sampling (SDS) (Poole et al., 2022) to perform editing by guiding the optimization of 3D representations with gradients derived from powerful pretrained diffusion models. Nowadays, the prevailing paradigm is to leverage diffusion models to generate or edit a sufficient number of views, which are then used to finetune the underlying 3DGS. However, while recent 2D diffusion models have seen significant breakthroughs in both generation quality and editing controllability, many of the latest 3D personalization approaches still rely on early U-Net-based architectures and approaches (Kim et al., 2024; Zhang et al., 2024), thereby failing to take advantage of the latest advancements in the image and video diffusion models. While a small number of recent approaches (Wang et al., 2025a; Zhang et al., 2025) have adopted state-of-the-art image and video diffusion models and demonstrated high-quality 3D editing results, they often depend on per-scene finetuning, which poses challenges in terms of efficiency and scalability. In contrast, our method not only fully leverages recent developments in 2D diffusion models (Labs et al., 2025; Wan et al., 2025), but also eliminates the need for per-scene training, achieving more compelling 3D editing in a simple yet elegant manner.

## 3 METHOD

We begin in Section 3.1 by elucidating TINKER's process for both few-shot and one-shot editing. We then provide a detailed account in Section 3.2 of how we constructed our dataset and model for multi-view consistent image editing. In Section 3.3, we introduce our model for scene completion from sparse views. Appendix C.2 and C.4 further discusses TINKER's potential for 4D Editing and additional applications.

### 3.1 3D EDITING WITH SPARSE VIEWS AS INPUT

**Editing with few-shot input.** As illustrated in Figure 3, given an original 3DGS $\mathcal{G}$, our objective is to generate an edited 3DGS $\mathcal{G}'$. We begin by rendering a few videos from $\mathcal{G}$ and randomly selecting a few sparse views. These selected views are edited using our multi-view consistent image editing model to produce the edited reference views. We then estimate the depth maps of the rendered video using Video Depth Anything (Chen et al., 2025). With these depth maps and the edited reference views, we employ our scene completion model to generate the images of the other views. Since the video is rendered from $\mathcal{G}$, we have access to the exact camera pose for each view, which allows the completed frames to be directly used in optimizing $\mathcal{G}$ into the edited $\mathcal{G}'$. The entire editing does not necessitate any per-scene finetuning.

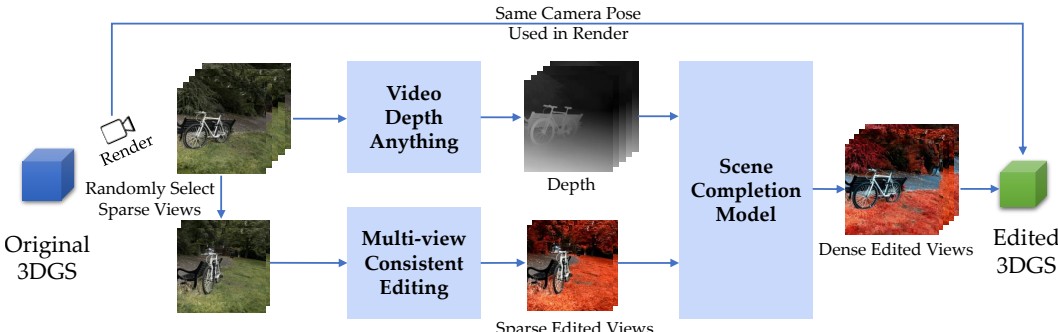

Figure 3: Overview of our editing process. We first apply our multi-view consistent editing model to obtain coherent sparse views. Leveraging depth constraints from the rendered results, we generate a large number of consistent edited images. The edited images are used to optimize the 3DGS to achieve high-quality 3D editing.

**Editing with one-shot input.** TINKER is also capable of handling a more challenging scenario: editing with one single reference view without additional training. The procedure remains identical to the few-shot scenario, with the key difference being that we select only a single sparse view from the rendered videos to serve as the initial reference for editing. This initial reference view is then used by the scene completion model to generate an initial set of edited views. These newly generated views, in turn, serve as subsequent reference views to progressively propagate the edit, with the process continuing until the entire scene is sufficiently covered by the generated views. As a result, edited $\mathcal{G}'$ is achieved by fine-tuning $\mathcal{G}$ with these generated views.

## 3.2 MULTI-VIEW CONSISTENT EDITING

Our approach begins with the observation that the state-of-the-art large-scale image editing model (Labs et al., 2025) is capable of achieving multi-view consistent editing when provided with two concatenated images as input. This simple strategy ensures consistency between the two concatenated views. However, it fails to enforce consistency across different image pairs, leading to global inconsistencies. A straightforward solution is to concatenate an edited image with an unedited one, prompting the model to apply similar edits to the latter by implicitly referencing the former. However, we find that this approach yields a very low success rate, with the model frequently reverting to simply reproducing the original unedited image. The model is not exposed to such reference-based editing configurations during its pretraining, and therefore lacks the knowledge needed to generalize in this setting.

Inspired by the strong local consistency exhibited by the large image editing model, we leverage the model itself to construct a large-scale reference-based editing dataset. Specifically, we begin by randomly selecting two different views of the same scene from publicly available 3D-aware datasets (Yeshwanth et al., 2023; Baruch et al., 2021; Liu et al., 2025; Ling et al., 2024; Xia et al., 2024; Zhou et al., 2018). Given a 3D-aware dataset $\mathbb{D} = \{\mathbf{I}_v^i\}$, where $\mathbf{I}_v^i$ denotes the $i$-th scene from view $v$, we sample 2 views $\mathbf{I}_a, \mathbf{I}_b \in \mathbb{D}$ of the same scene from different viewpoints. We omit the scene index $i$ for simplicity to indicate that 2 views are sampled from the same scene. Subsequently, we generate editing prompts $P$ with a large language model and perform editing using model $\mathcal{E}$:

$$\mathbf{I}_a', \mathbf{I}_b' = \mathcal{E}(\text{Concat}(\mathbf{I}_a, \mathbf{I}_b), P) \tag{1}$$

To ensure that the editing is effectively applied, we compute the feature similarity between the edited and original images using DINOv2 (Oquab et al., 2023):

$$s_1 = \text{sim}(f_{dino}(\mathbf{I}_a), f_{dino}(\mathbf{I}_a')), s_2 = \text{sim}(f_{dino}(\mathbf{I}_b), f_{dino}(\mathbf{I}_b')), s_{noedit} = \max(s_1, s_2) \tag{2}$$

We discard pairs with overly high similarity scores that exceed a threshold $\tau_{noedit}$ by $s_{dino} > \tau_{noedit}$, indicating insufficient editing. Furthermore, we evaluate the similarity between the two edited views by $s_{mv} = \text{sim}(f_{dino}(\mathbf{I}_a'), f_{dino}(\mathbf{I}_b'))$, and filter out samples with low inter-view consistency below a threshold $\tau_{mv}$ by $s_{mv} < \tau_{mv}$. Finally, we construct training inputs by pairing an original image with a reference edited image from a different view, and fine-tune the model using LoRA (Hu et al., 2021) to learn reference-based editing. This process empowers the model to

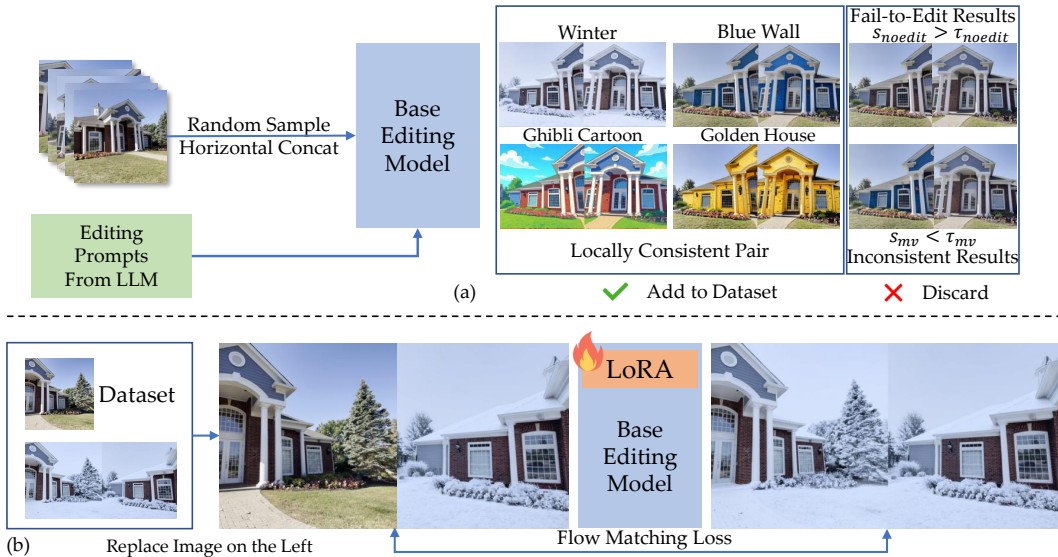

Figure 4: (a) We leverage recent base editing model to generate a large number of consistent image pairs and discard the inconsistent and fail-to-edit results. (b) The data generated in (a) is used to fine-tune the base editing model itself. Specifically, we horizontally concatenate the input image and an edited image, and train the model using LoRA to learn how to perform referring-based editing.

generalize edits across views and achieve globally consistent results. This process and the training objective can be formulated as:

$$\mathbf{I} = \text{Concat}(\mathbf{I}_a, \mathbf{I}'_b), \mathbf{I}' = \text{Concat}(\mathbf{I}'_a, \mathbf{I}'_b), \mathbf{z}_0 = g(\mathbf{I}), \mathbf{z}' = g(\mathbf{I}')$$

$$\text{Loss} = \mathbb{E}_{\mathbf{z}_0, t} \| \mathcal{E}_\theta(\mathbf{z}_t, t, P) - u(\mathbf{z}'_t) \|_2^2,$$

(3)

where $g$ is the Variational Autoencoder that maps images to the latent space. We employ a flow matching loss (Liu et al., 2022; Lipman et al., 2022; Albergo & Vanden-Eijnden, 2022; Esser et al., 2024) to minimize the discrepancy between the model's predicted velocities and the ground truth velocities $u$.

## 3.3 SCENE COMPLETION MODEL

While it is feasible to perform view-by-view editing by leveraging our multi-view consistent image editing model using sparse reference views, this approach is extremely time-consuming. Motivated by the recent success of video diffusion models, we aim to exploit their strong spatial-temporal priors to achieve efficient scene completion with sparse edited views. A natural idea is to train a model that edits the original 3DGS rendered video into a target video, conditioned on multi-view consistent edited images. However, there are no such editing datasets currently available. **Therefore, we rethink the problem by casting the training objective of the editing task into the reconstruction task.** Specifically, we aim to train a model to reconstruct the original scene from sparse views, with the goal of generalizing to reconstructing the edited scene from edited views, thus achieving editing through reconstruction.

In generative modeling, several works have attempted to condition multi-view generation on ray maps that encode camera parameters. However, we find that this approach lacks sufficient geometric constraints, often resulting in view inconsistencies. Upon further analysis, we argue that depth is a more suitable conditioning signal in the context of 3D. Depth not only explicitly encodes structural constraints but also implicitly carries information about camera pose. Moreover, it greatly benefits from recent advances in depth estimation.

Recent unified model for video generation and editing, VACE (Jiang et al., 2025), has explored depth-guided video editing. However, as generation is their main objective, they treat depth more as a soft reference than an explicit constraint, resulting in results that deviate from the intended geometric structure. Although its performance is remarkable in depth-guided generation, this flexibility

is undesirable in 3D editing scenarios where we require precise control over specific areas while maintaining geometric consistency in unedited regions. 3D editing necessitates a 3D-aware scene completion model that accurately follows depth constraints to maintain strict geometric consistency throughout the scene.

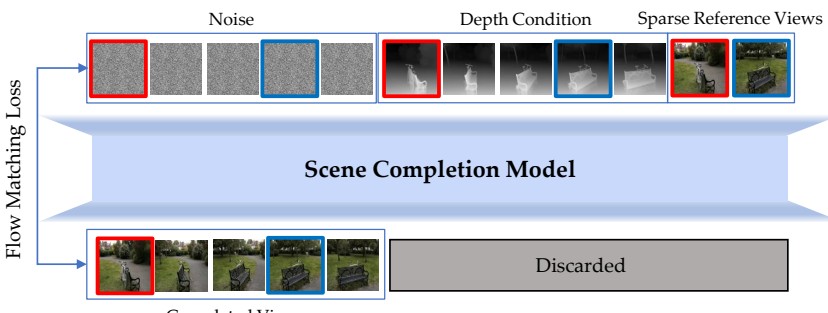

Figure 5: Visualization of our scene completion model. Contours of the same color denote elements that share the same positional embedding. During training, the flow matching loss is computed exclusively between the model outputs corresponding to the noisy latents and the ground-truth video frames. Outputs corresponding to the conditioning tokens are discarded and do not contribute to the loss calculation.

Based on the above analysis, we develop our scene completion model tailored for 3D. To this end, we leverage the pretrained WAN2.1 (Wan et al., 2025) to train an image-to-video model that accurately follows depth conditions. Given the scarcity of 3D video datasets, we take inspiration from Diception (Zhao et al., 2025), which achieves strong performance with limited data. Specifically, we treat the depth maps as RGB images and process them into tokens, following the original procedure of Wan2.1. The reference views are also processed into tokens in the same manner. The noisy latent tokens $\mathbf{Z}^t = [\mathbf{Z}_0^t, \mathbf{Z}_1^t, \ldots, \mathbf{Z}_N^t]$ at timestep $t$, depth tokens $\mathbf{D} = [\mathbf{D}_0, \mathbf{D}_1, \ldots, \mathbf{D}_N]$, and reference view tokens $\mathbf{V}$ are concatenated along the sequence dimension to form the model input. The training process also follows the flow matching, which is formulated as:

$$
\mathbf{X}_{input}^t = \text{Concat}(\mathbf{Z}^t, \mathbf{D}, \mathbf{V})
$$
$$
\text{Loss} = \mathbb{E}_{\mathbf{z}_0, t}\|\Phi_\theta(\mathbf{X}_{input}^t, t) - u(\mathbf{Z}^t)\|_2^2. \tag{4}
$$

During training, we always provide the first frame as the default reference view, and randomly select 0 to 2 additional reference views to help the model learn both one-shot and few-shot scene completion. The text embedding is fixed to a constant embedding, enhancing the model to focus solely on depth-guided generation. To enable the model to associate these reference views with the target $j$-th frame, we assign them the same positional embedding as the target frames:

$$
\text{PE}(\mathbf{V}) = \text{PE}(\mathbf{D}_j) = \text{PE}(\mathbf{X}_j). \tag{5}
$$

Through this design, the model effectively learns which regions of the scene correspond to the reference view, and is able to leverage both depth and reference views to achieve high-quality scene completion.

## 4 EXPERIMENTS

### 4.1 COMPARATIVE EXPERIMENTS

Due to space limitations, the implementation details are provided in Appendix B. Building upon this implementation, we conduct a comprehensive qualitative and quantitative comparison with the latest high-quality 3D editing methods (Wu et al., 2024b; Chen et al., 2024a; Zhuang et al., 2024; Lee et al., 2025) in terms of both output quality and computational cost. Our evaluation is conducted on the Mip-NeRF-360 (Barron et al., 2022) and IN2N (Haque et al., 2023) datasets. For each scene, we perform editing using the same textual prompt, and subsequently render the edited scenes from uniformly sampled camera poses to obtain results from different methods under identical novel viewpoints. For methods necessitating image input, we follow their official preprocessing pipelines

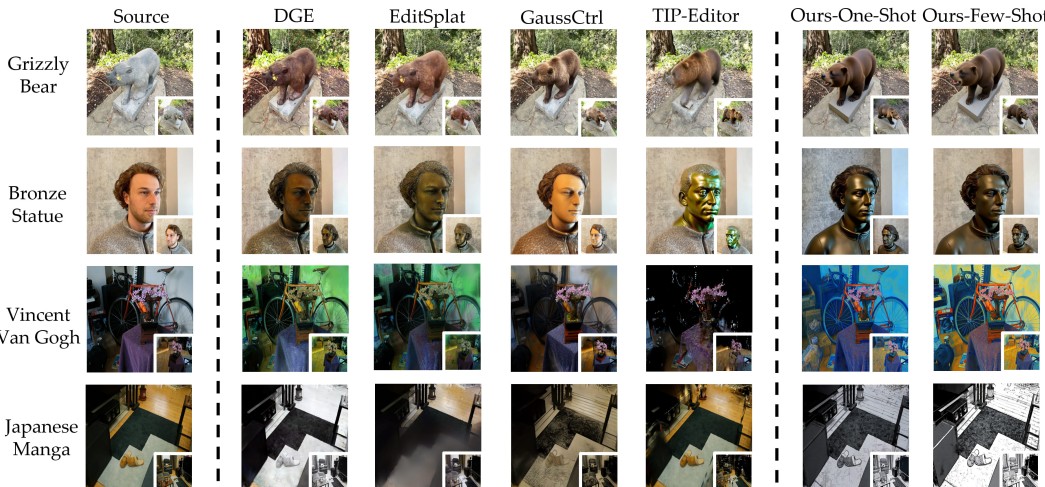

Figure 6: Qualitative comparisons of novel views in different methods. Our approach demonstrates both better visual quality and more diverse editing results.

Table 1: Quantitative comparisons of different methods. TINKER achieves superior results with acceptable computational cost.

| | Quality | | | Computational Cost | |
|---|---|---|---|---|---|
| | CLIP-dir↑ | DINO↑ | Aesthetic↑ | On 24G GPU | Avg. Editing Time↓ |
| DGE | 0.102 | 0.948 | 5.747 | **Yes** | **10min** |
| GaussCtrl | 0.123 | 0.957 | 5.624 | No | 20min |
| TIP-Editor | 0.084 | 0.875 | 5.397 | No | 35min |
| EditSplat | 0.102 | 0.956 | 5.661 | **Yes** | 19min |
| Ours-one-shot | 0.143 | 0.958 | 6.214 | **Yes** | 15min |
| Ours-few-shot | **0.157** | **0.959** | **6.338** | **Yes** | 15min |

to prepare the corresponding images. We use NeRFStudio (Tancik et al., 2023) for 3DGS optimization and rendering. Furthermore, we quantitatively evaluate the results using four metrics: CLIP Text-Image directional similarity (Radford et al., 2021) used in GaussCtrl (Wu et al., 2024b) assess semantic alignment, DINO similarity (Oquab et al., 2023) between the edited renderings to measure cross-view consistency, and aesthetic score (Schuhmann et al., 2022) to assess rendering quality.

As demonstrated in Figure 6 and Table 1, in both one-shot and few-shot settings, our approach consistently outperforms existing methods for both object-level and scene-level editing. Furthermore, some methods, such as TIP-Editor (Zhuang et al., 2024), require per-scene fine-tuning, making them infeasible to run on consumer-grade 24 GPUs, whereas our method works entirely without further per-scene fine-tuning and can be executed efficiently on a single consumer-grade GPU. Moreover, we observe that while some approaches such as TIP-Editor (Zhuang et al., 2024) are capable of producing high-quality object-level edits, they fall short in performing scene-level edits. In contrast, our method supports both object-level and scene-level 3D editing with higher quality, even for scenes with substantial overall style changes, such as oil paintings or black-and-white comics.

## 4.2 ABLATIONS AND ANALYSES

We performed a comprehensive set of ablation studies and analyses to better understand the effectiveness of our design choices. Specifically, we examined the effectiveness of fine-tuning our multi-view consistent editing model, assessed the effect of concatenating additional images for consistent editing, and evaluated the advantages of employing depth as a conditioning signal over the ray-map conditioning used in prior work. Moreover, we analyzed the strengths of our approach relative to existing depth-guided video generation methods. *Owing to space constraints, comprehensive additional results, 4D editing results, video-related ablation studies, and an detailed exploration of further applications, are presented in Appendix C.1, C.2, C.3, C.4 respectively.* Taken together, our

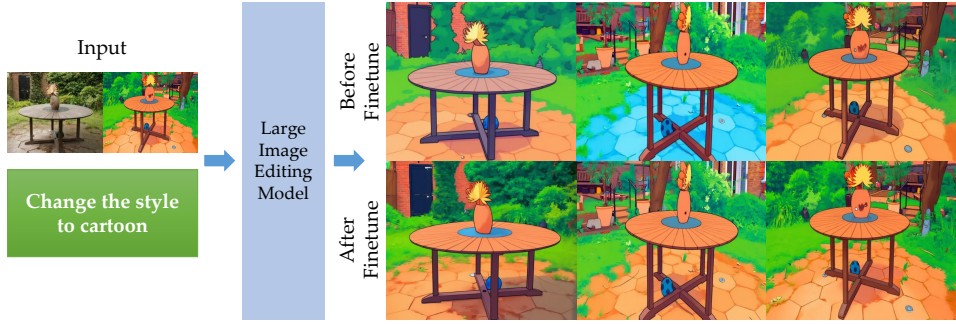

Figure 7: Qualitative comparisons before and after multi-view consistent image editing fine-tuning. After fine-tuning, our model can perform edits guided by the provided reference image, effectively ensuring the global consistency.

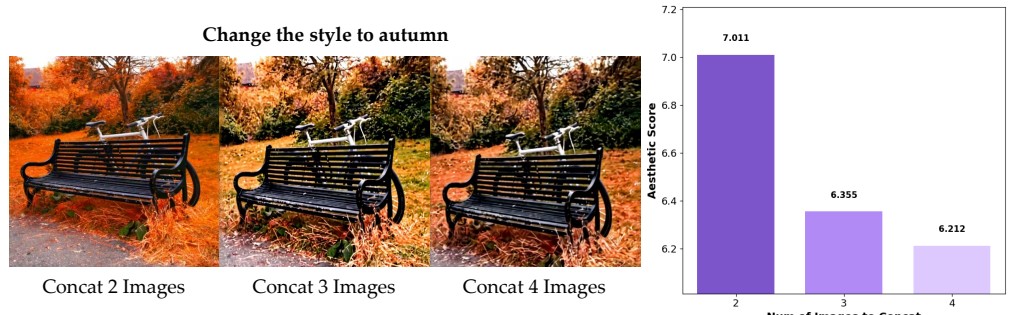

Figure 8: Effect of the number of horizontally concatenated images on visual quality. Concatenating too many images leads to a significant degradation in image quality, while concatenating two images yields the best results.

comprehensive experiments not only validate the effectiveness of TINKER's core design choices but also demonstrate its remarkable capabilities.

**Multi-view consistent editing finetuning.** To validate the effectiveness of our fine-tuning for multi-view consistent editing, we compare the global consistency of the results and the model's ability to align with the provided reference images before and after fine-tuning. We use Mip-NeRF 360 (Barron et al., 2022) as the evaluation set, where 10 prompts are applied to each scene for referring editing, producing 20 images per prompt per scene. We then compute the DINO similarity (Oquab et al., 2023) among the generated images for each prompt in each scene and take the average across all scenes, which serves as a measure of cross-view consistency. As shown in Figure 7 and Table 2, fine-tuning significantly improves both the global consistency of the edited results and the model's ability to faithfully follow the editing cues from the reference views.

**Concatenating more images for multi-view consistent editing.** An intuitive approach is to concatenate all multi-view images intended for editing and feed them into the model to achieve consistent editing. However, we observed that this strategy leads to poor results and multi-view inconsistency. Even under a minimal setting, editing with just three concatenated images, the output images exhibit noticeable quality degradation. This issue arises because the underlying base model imposes a constraint on the image resolution. As a result, input images are automatically resized to fit within the limit. When too many images are concatenated, each individual image is heavily downsampled, leading to substantial loss of detail and visible blurring. As shown in Figure 8, we compare the Aesthetic Score across different numbers of concatenated images and find that concatenating two images strikes a reasonable balance between consistency and visual fidelity.

Table 2: After multi-view consistent image editing fine-tuning, the edited images exhibit substantially improved multi-view consistency, while maintaining comparable text–image alignment and aesthetic quality to the non-finetuned results.

|          | Before | After |
|----------|--------|-------|
| DINO     | 0.862  | **0.943** |
| CLIP-dir | 0.277  | **0.281** |
| Aesthetic| **7.058** | 6.973 |

## 5    CONCLUSION

We propose **TINKER**, to the best of our knowledge, the first general-purpose 3D editing framework that eliminates the need for per-scene optimization. TINKER bridges a critical gap by extending the breakthroughs of 2D diffusion models into the domain of 3D editing, enabling high-quality results in few-shot even one-shot settings. We also introduce the first dataset and data pipeline specifically designed for multi-view consistent editing to benefit future researches. Beyond editing, TINKER also demonstrates additional versatility across tasks such as video compression and video editing, showcasing the potential of a unified 2D, 3D, even 4D editing framework. We believe that TINKER offers a scalable, flexible, and generalizable solution for future editing research.

## ACKNOWLEDGMENTS

This work was in part supported by the National Key R&D Program of China (No. 2022ZD01601-60101), Ningbo Science and Technology Bureau (No. 2024Z291) and the National Natural Science Foundation of China (No. 62206244).

## ETHIC STATEMENT

This work adheres to the ICLR Code of Ethics. The research presented in this paper is based on publicly available datasets and models, and we focus on algorithmic advancements. We have analyzed the potential risks and societal impacts of our work and have not identified any significant ethical concerns. Our study does not involve sensitive data or applications with a high potential for harm. We are committed to the responsible development and application of machine learning.

## REPRODUCIBILITY STATEMENT

To ensure the reproducibility of our research, we have made the following resources available:

**Implementation Details:** A detailed description of our method and the formulation of the loss functions are detailed in Section 3. The hardware, open-source models, hyperparameters and experimental settings required to reproduce our main results are detailed in Appendix B. This includes learning rates, batch sizes, model architectures, and the specific computational infrastructure used (e.g., GPU type, number of GPUs).

**Data:** All datasets used in this study are publicly available. In Appendix B, we provide detailed descriptions of the datasets, along with preprocessing steps and links to the original sources.

**Code Availability:** The source code and pre-trained models for this paper will be made publicly available upon acceptance to facilitate reproducibility and encourage future research.

We believe that these resources provide a comprehensive basis for the research community to reproduce and build upon our work.

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

## THE USE OF LARGE LANGUAGE MODELS

We utilized the Large Language Model (LLM) for two specific and limited purposes. First, it was employed as a writing assistance tool to improve the grammar, clarity, and readability of the manuscript. Second, it was used to generate the instructional text for our referring editing dataset. LLM did not contribute to the formulation of core research ideas, the development of the methodology, or experimental results and conclusions presented in this paper.

## A    APPENDIX OVERVIEW

This appendix presents supplementary materials to complement the main paper, offering further implementation details, additional results, and extended ablations and analyses. The contents are organized as follows:

- **Appendix B: Implementation Details**

    A detailed breakdown of our implementation, covering:

    - B.1: Training details and dataset specifics for the multi-view consistent editing model, including key hyperparameters and GPU configurations
    - B.2: Training details and dataset specifics for the scene completion model.

- **Appendix C: Additional Results and Analyses**

    A collection of further results and in-depth analyses, including:

    - C.1: Additional visualizations of our 3D editing results.
    - C.2: Showcase of our 4D dynamic scene editing capabilities.
    - C.3: Further ablation studies and analyses validating the design choices of TINKER.
    - C.4: Demonstrations of other applications, such as quality refinement, video reconstruction, and test-time optimization.

- **Appendix D: Limitations and Discussions**

    A discussion of the potential limitations of our method.

## B    IMPLEMENTATION DETAILS

### B.1    MULTI-VIEW CONSISTENT IMAGE EDITING DATASET AND MODEL

For the multi-view consistent image editing model, we adopt Flux Kontext (Labs et al., 2025) as the foundation model and construct a referring multi-view consistent image editing dataset following the procedure detailed in Section 3.2. We use GPT-o3 to generate 400 image editing instructions. The inputs of our data pipeline are sourced from publicly available 3D-aware datasets, including DL3DV (Ling et al., 2024), WildRGBD (Xia et al., 2024), and uCO3D (Liu et al., 2025). We randomly select two images from each scene, concatenate them, and perform editing. The results are filtered using the procedure described in Section 3.2 to determine whether they should be retained. During dataset construction, two thresholds for data validity, $\tau_{noedit}$ and $\tau_{mv}$, are set quite strictly. Although this may occasionally filter out some good samples, it effectively reduces the number of bad cases in the dataset, thereby benefiting model training. Specifically, we set $\tau_{noedit} = 0.95$ and $\tau_{mv} = 0.9$ to ensure sufficient data quality. In total, our dataset comprises 250,000 samples, each containing two original images, one edited image, and the corresponding editing instruction. We show the input to Large Vision Language Model to generate the editing prompts in Figure S12 and illustrate some data samples in Figure S13. Both the dataset and the data generation pipeline will be released to facilitate further research in this area.

When fine-tuning for referring editing using LoRA (Hu et al., 2021), we apply LoRA with rank 128 to all the query, key, value, and output layers of the base model. Training is performed with a dropout rate of 0.05 for 30,000 iterations on four NVIDIA H100 GPUs, using a constant learning rate of 2e-5 and the AdamW optimizer (Loshchilov & Hutter, 2017).

## B.2 SCENE COMPLETION MODEL

For scene completion model, we employ Wan2.1 1.3B model (Wan et al., 2025) as the foundational backbone of our scene completion model. Our model undergoes a two-stage training protocol. Initially, it is pre-trained on the large-scale OpenVid-1M dataset (Nan et al., 2024). Subsequently, to instill robust 3D-aware capabilities, we fine-tuned the model on a curated collection of 3D-centric datasets, including DL3DV (Ling et al., 2024), Re10k (Zhou et al., 2018), ArkitScenes (Baruch et al., 2021), WildRGBD (Xia et al., 2024), and uCO3D (Liu et al., 2025). Depth annotations for our training data are generated using the Video Depth Anything model (Chen et al., 2025). The training of scene completion model was conducted for 200,000 iterations on a cluster of 16 NVIDIA H100 GPUs using a constant learning rate of $2e-5$. The evaluation datasets are from Mip-NeRF-360 (Barron et al., 2022) and IN2N (Haque et al., 2023). Finally, we use NeRFStudio (Tancik et al., 2023) for 3DGS optimization and rendering.

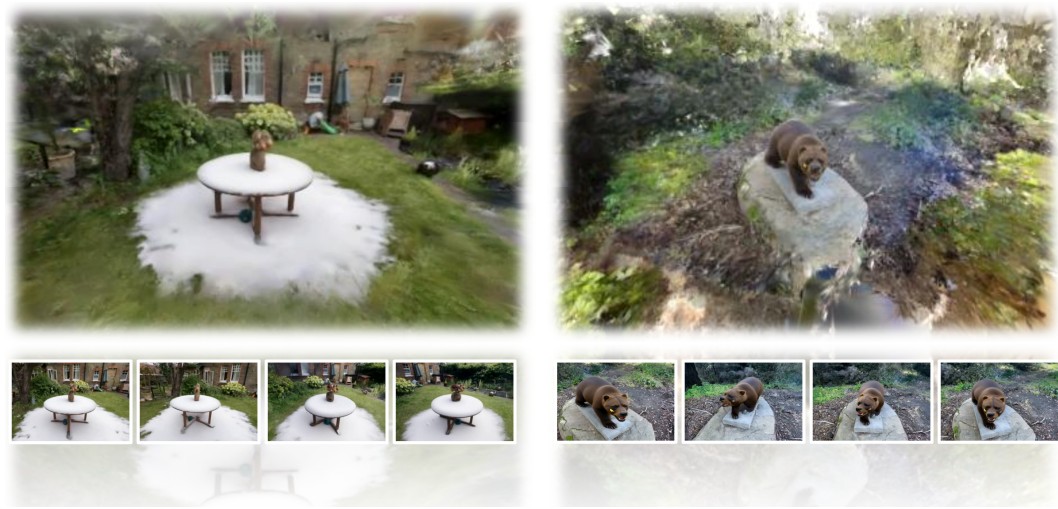

Figure S1: Visualizations of edited 3DGS and renderings using NeRFStudio.

## C ADDITIONAL RESULTS

### C.1 ADDITIONAL VISUALIZATIONS

We employ NeRFStudio (Tancik et al., 2023) to optimize the 3D Gaussian Splatting using the edited views as input, as illustrated in Figure S1. In this section, we further present additional one-shot and few-shot 3D editing results in Figure S8, S9, S10, S11. These comprehensive visualizations demonstrate that our method significantly lowers the usage barrier, enabling high-quality object-level and scene-level 3D editing of various styles without requiring per-scene fine-tuning.

### C.2 TINKER FOR 4D EDITING

Beyond static scenes, TINKER's capabilities extend compellingly into the temporal domain, showcasing a strong potential for 4D dynamic scene editing. We validate this on the challenging scenes from DyNeRF dataset (Li et al., 2022), leveraging 4D Gaussian Splatting (4DGS) (Wu et al., 2024a) as the underlying dynamic scene representation. Crucially, our editing pipeline for 4D scenes remains the same as our editing pipeline for 3D scenes. The process begins with our sparse multi-view consistent editing technique to generate a small set of edited, yet consistent, anchor views. Subsequently, our scene completion model utilizes these sparse views to generate dense and temporally coherent images, which are then used to optimize the downstream 4DGS model, effectively propagating the desired edit throughout the entire dynamic sequence.

As illustrated in Figure S2, a key strength of our method is its versatility, enabling both localized, object-level modifications and global, scene-level stylizations within the 4D context. TINKER can seamlessly alter the appearance of a specific object or transform the entire scene's atmosphere over time. Across all visualized scenarios, our framework consistently produces high-quality results. The

generated edits are not only visually striking in individual frames but also maintain strict temporal coherence, ensuring smooth and consistent dynamics, with the final results appearing natural and harmonious within the original scene, free of any jarring or obtrusive artifacts.

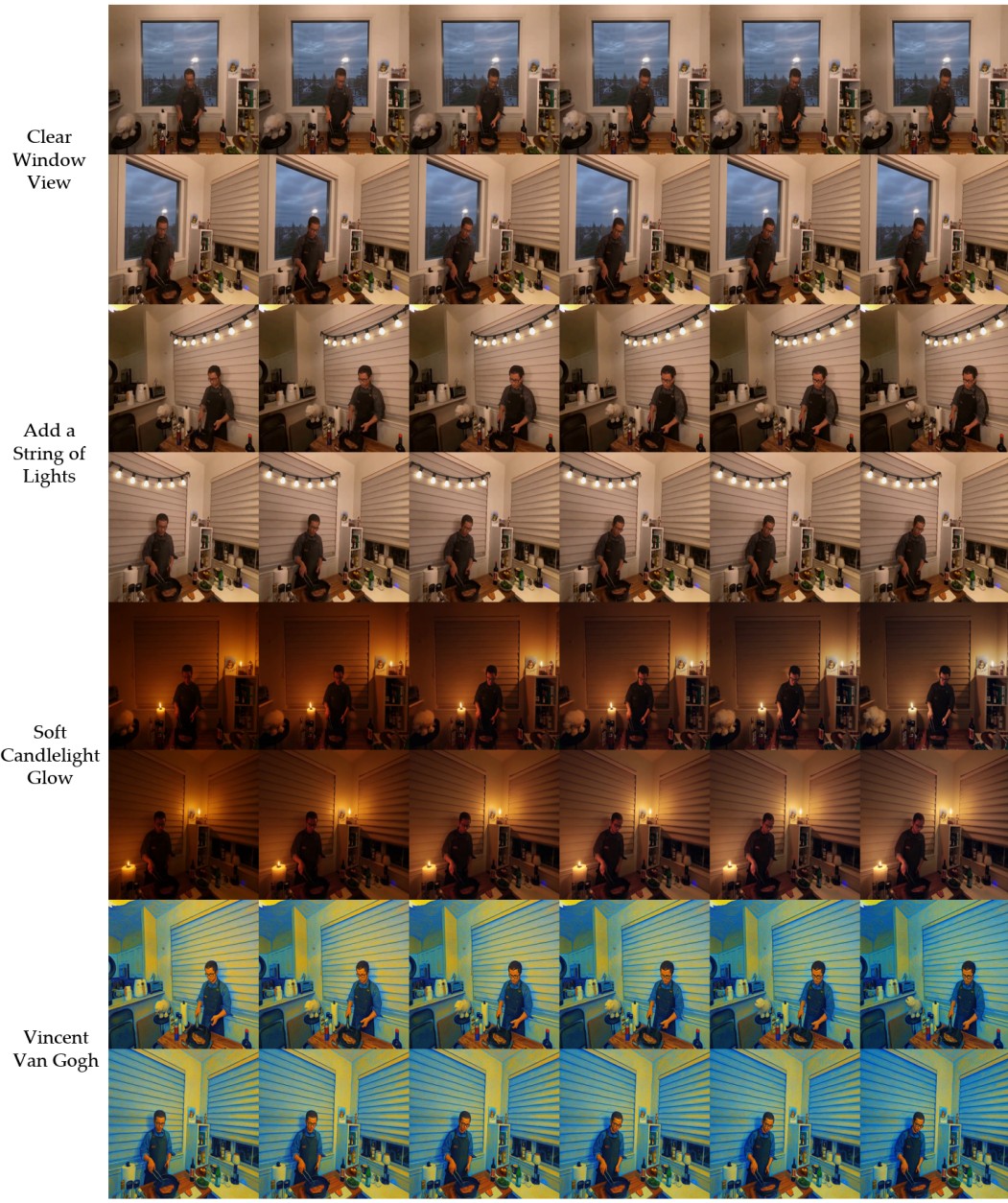

Figure S2: Visualizations of 4D dynamic scene editing with TINKER. Our results on DyNeRF sequences showcase both object-level and scene-level edits, demonstrating high-fidelity, harmonious, and temporally coherent results.

## C.3 ADDITIONAL ABLATIONS

**Advantages of depth as condition in scene completion.** Existing methods predominantly rely on diffusion models to generate new views. Some approaches (Gao et al., 2024) condition on ray maps to generate missing views, while others (Wang et al., 2024b) directly interpolate between the first and last frames using diffusion models. We systematically compare our scene completion model with both types of methods to demonstrate its superiority. First, we train a scene completion model

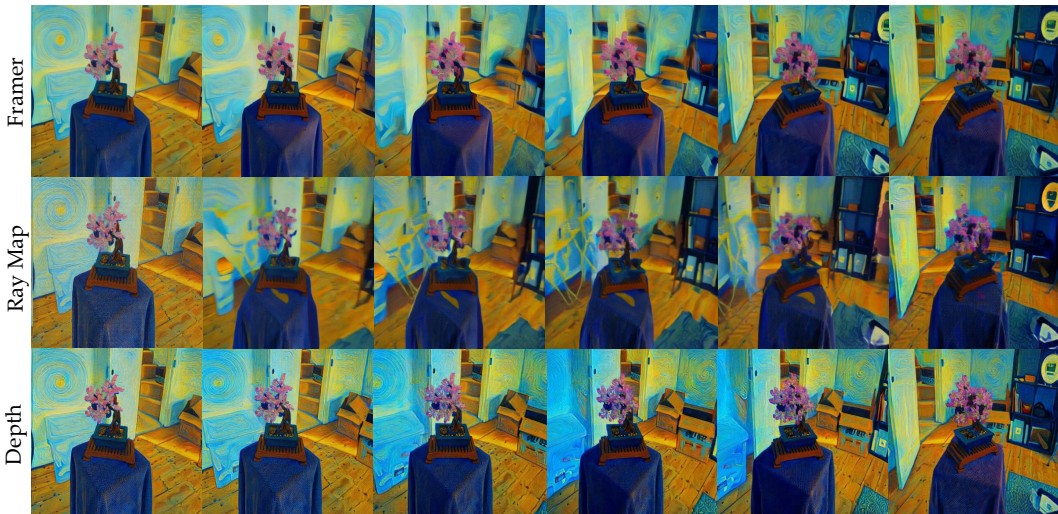

Figure S3: Qualitative comparisons of different scene completion methods. Conditioning on depth produces results with one-to-one corresponding camera poses, while achieving superior geometry and detail preservation without being restricted to first and last frame as inputs.

conditioned on ray maps in a manner similar to prior work (Gao et al., 2024). We observe that, due to the lack of explicit geometric constraints, this approach often results in noticeable geometric distortions in the generated views. Moreover, it often generates outputs that violate the constraints of the camera ray map, or inconsistent results with original scenes. As for the second approach, interpolating between the first and last frames suffers from both visual artifacts and a lack of camera pose information for the generated intermediate views, making them unsuitable for downstream 3DGS optimization. Moreover, this strategy imposes a rigid constraint on input format, limiting the input to the first and last frames and thereby reducing flexibility in editing scenarios. In contrast, our method leverages depth to provide strong geometric guidance and maintain tight alignment with the corresponding camera poses. It also supports arbitrary reference views, not limited only to the first and last frames. As evidenced by the results in Figure S3 and Table S1, our method significantly outperforms the aforementioned baselines in both quality and consistency.

**Advantages over existing depth-conditioned diffusion models.** Several recent works, such as VACE (Jiang et al., 2025), have explored using depth as condition for video generation. However, these methods are typically trained on natural video datasets and do not pay much attention to 3D-related data. Furthermore, these methods treat the depth condition as a reference rather than a constraint that must be strictly enforced. Consequently, while they are capable of producing high-quality depth-guided video generation, the resulting outputs often fail to strictly adhere to the provided depth constraints, which is not desirable in 3D settings where geometric consistency is essential. We compare our approach with the latest existing depth-guided video generation methods VACE (Jiang et al., 2025) in Figure S4 and Table S1, which shows that our model better understands camera motion and more faithfully respects depth constraints. In addition to supporting depth-guided video generation, VACE also allows controlling the editing region via masks. We compare our method against both of these capabilities. For depth-guided video generation, the results exhibit clear multi-view inconsistencies. For mask-based editing, the results also show certain fine-grained multi-view inconsistencies, and the quality preservation in detailed regions is significantly lower than that achieved by our method. We attribute this success to removing the text prompt input and training the model on 3D-aware datasets to strictly follow the provided depth.

## C.4 APPLICATIONS

**Quality refinement.** We observe that our model can effectively enhance the quality of rendered results, as this refinement can also be regarded as a special variant of editing. As shown in Figure S5, by employing a prompt such as "enhance the quality," we can guide the model to refine blurry areas in the rendering, yielding outputs with sharper details. This refinement process allows our method to improve the overall fidelity of the 3DGS reconstruction.

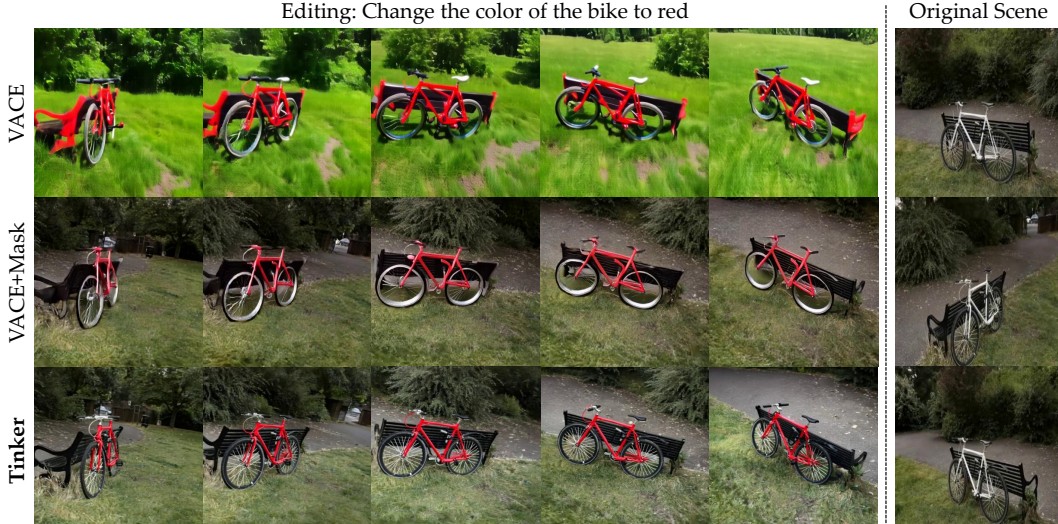

Figure S4: Comparison with VACE in both depth-guided video generation and mask-based editing. Our method demonstrates superior multi-view consistency and better preservation of fine details.

Table S1: Quantitative comparisons of different conditions and different depth-guided video generation models. Our approach achieves the best overall performance.

|  | Text-Image Similarity↑ | DINO↑ | Aesthetic↑ |
|---|---|---|---|
| VACE | 0.760 | 0.916 | 5.833 |
| VACE+Mask | 0.799 | 0.954 | 6.118 |
| Framer | 0.773 | 0.973 | 6.227 |
| Ours-Ray-Map | 0.783 | 0.931 | 6.214 |
| Ours-Depth | **0.821** | **0.978** | **6.586** |

**Video reconstruction.** TINKER reconstructs high-quality videos from just the first frame and the corresponding depth sequence. As shown in Figure S6 and Table S2, our approach achieves temporally coherent reconstructions with sharp details and faithful geometry, demonstrating its effectiveness across both qualitative and quantitative evaluations. The evaluation dataset consists of 1,000 videos sampled from OpenVid-1M (Nan et al., 2024), and we ensure these videos for evaluation are not included in our training set. We observe that the latest model (Jiang et al., 2025) taking the first frame and depth as inputs, while capable of producing high-quality video generation, fails

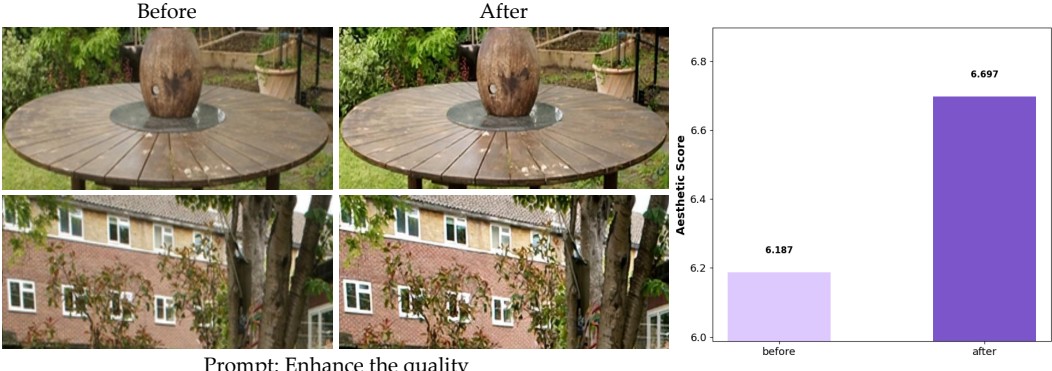

Prompt: Enhance the quality

Figure S5: TINKER demonstrates the ability to refine blurry regions, recovering sharper structures and finer details while maintaining overall visual consistency, as this refinement can be regarded as a special type of editing.

to accurately reconstruct the original video content. In contrast, our method significantly improves reconstruction accuracy, achieving faithful recovery of both geometric structure and appearance. Furthermore, because the model operates directly on grayscale depth maps, it naturally supports a compact video representation in which an entire video can be stored as its grayscale depth sequence and a single first frame. This property highlights the potential of our method not only for high-fidelity video reconstruction but also for efficient video compression and storage.

Table S2: Quantitative comparisons of video reconstruction with first frame and depth as input.

|         | PSNR↑   | SSIM↑ |
|---------|---------|-------|
| VACE    | 16.635  | 0.331 |
| TINKER  | **31.869** | **0.941** |

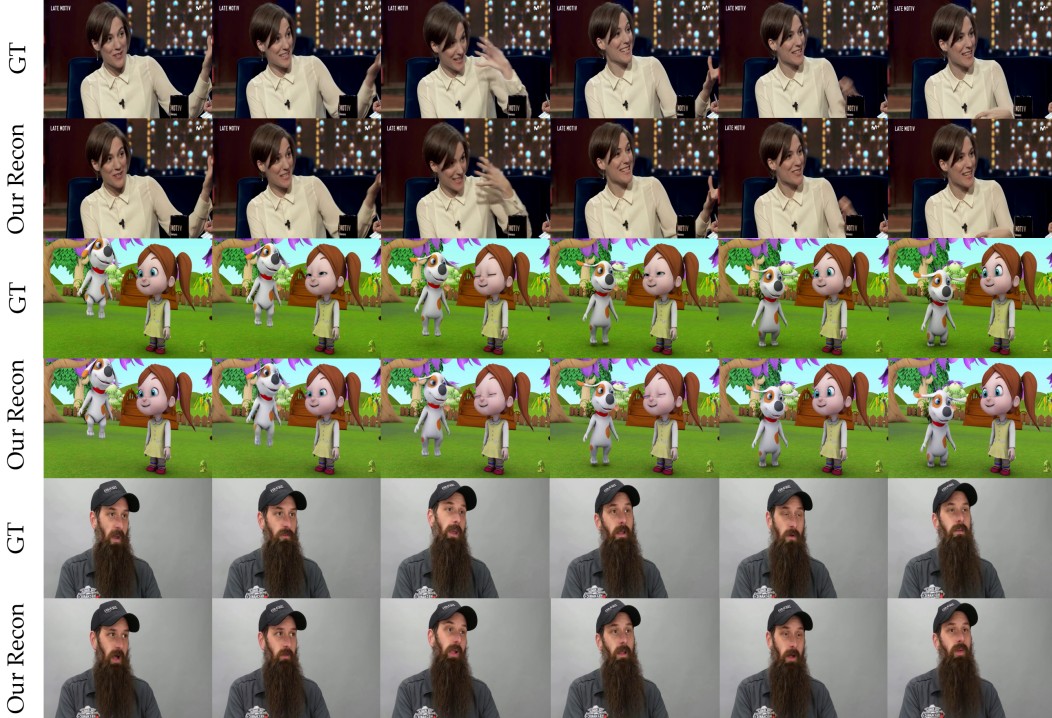

Figure S6: TINKER demonstrates the capability of high-quality video reconstruction with only the first frame and depth maps as input.

**Test-time optimization.** Most importantly, since our method does not require per-scene finetuning, it supports test-time optimization: users can iteratively experiment with different reference views, replacing generated views of low quality in the last generation process with newly generated ones using the scene completion model. This iterative process leads to higher-quality 3D editing results.

### C.5 BASELINES WITH FLUX

To rigorously isolate the contributions of our TINKER pipeline, we investigate whether prior baseline methods could achieve comparable results simply by substituting their U-Net-based 2D editors with the more advanced FLUX model. We implemented and evaluated FLUX-adapted versions of baselines, concluding that such a naive substitution is unviable.

Specifically, the FLUX-adapted methods we implement include:

- Instruct-GS2GS-FLUX: An adaptation of iterative dataset optimization methods similar to Instruct-NeRF2NeRF.

- DGE-FLUX and GaussCtrl-FLUX: Adaptations of methods reliant on multi-view feature alignment, such as Feature Injection or Feature Alignment. As FLUX is a flow-based architecture, we replace operations requiring DDIM Inversion with the compatible RF-Solver where necessary.

Our experiments revealed that simply replacing the editor not only failed to produce significant improvements but also introduced critical failures and prohibitive costs, as shown in Table S3 and Figure S7.

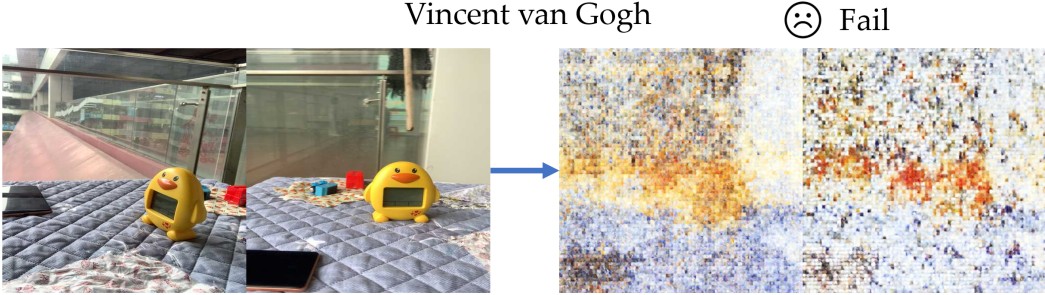

Figure S7: Prior methods' attention designs are tightly coupled with the U-Net architecture, causing severe performance degradation when the editing model is directly replaced.

Table S3: Quantitative comparisons of FLUX-adapted different methods. Simply replacing U-Net editors with FLUX is unviable, leading to prohibitive costs and even failed due to critical architectural mismatches.

| | Quality | | | Computational Cost | |
|---|---|---|---|---|---|
| | CLIP-dir↑ | DINO↑ | Aesthetic↑ | On 24G GPU | Avg. Editing Time↓ |
| DGE-FLUX | FAIL | FAIL | FAIL | No | FAIL |
| GaussCtrl-FLUX | FAIL | FAIL | FAIL | No | FAIL |
| Instruct-GS2GS-FLUX | 0.107 | 0.932 | 6.082 | No | 133min |
| Ours-one-shot | 0.143 | 0.958 | 6.214 | **Yes** | 15min |
| Ours-few-shot | **0.157** | **0.959** | **6.338** | **Yes** | 15min |

**Prohibitive computational cost.** For iterative methods like Instruct-GS2GS-FLUX, the computational overhead became impractical. The high intrinsic inference cost of the large FLUX model, when compounded by the numerous dataset refinement iterations required by the Instruct-NeRF2NeRF paradigm, results in an extreme computational burden that renders the approach unviable for practical application.

**Critical problems from different architectural design.** A more fundamental failure occurred in methods like DGE-FLUX and GaussCtrl-FLUX. These methods are foundationally dependent on multi-view feature alignment (e.g., DGE's STAttn, GaussCtrl's attention alignment), which fails due to an architectural incompatibility between U-Net and DiT. In U-Net, attention features generally do not contain dominant, explicit positional encoding. This allows cross-view attention mechanisms to match features based purely on content similarity. For example, an object patch at position $p_1$ in View 1 can correctly attend to the same object patch at position $p_2$ in View 2. In DiT, positional encoding is deeply integrated into the attention blocks. When naively adapting the baselines, the introduction of the DiT's strong PE fundamentally disrupts the model's ability to understand semantic content during multi-view alignment with prior attention designs. This prevents the model from correctly aligning the object, causing the multi-view feature alignment and subsequent editing to fail.

In conclusion, the attention designs of prior methods are architecturally coupled to the U-Net design, and their core mechanisms cannot be trivially upgraded by substituting the 2D editor. Conversely,

TINKER demonstrates superior quality and flexibility, supporting 2D, 3D, and 4D editing, which traditional methods are unable to achieve.

## C.6 USER STUDY

To quantitatively assess the subjective quality of our editing results, we further conduct a comprehensive user study. We evaluate user preferences along three key dimensions: text similarity (how well the edited scene matches the semantic meaning of the target text prompt), editing quality (the visual fidelity, realism, and coherence of the edited region), and multi-view consistency. Participants are asked to rate each result on a scale from 1 to 5, where higher scores indicate better results. In total, we collected 50 evaluation user studies where participants scored 20 different editing results. As shown in Table S4, the results clearly indicate that our method is strongly favored by users and better aligns with their subjective preferences compared to baseline approaches.

Table S4:    We conducted a user study across three dimensions: text similarity, editing quality, and multi-view consistency. The results indicate that our method better aligns with subjective user preferences.

|  | Text Similarity↑ | Editing Quality↑ | Multi-view Consistency↑ |
|---|---|---|---|
| GaussCtrl | 3.62 | 3.79 | 3.87 |
| DGE | 3.78 | 3.44 | 3.73 |
| EditSplat | 3.17 | 2.92 | 3.42 |
| TIP-Editor | 2.57 | 2.89 | 3.56 |
| Ours-OneShot | 4.38 | 4.49 | 4.31 |
| Ours-FewShot | **4.52** | **4.61** | **4.55** |

## D    LIMITATIONS AND DISCUSSIONS

Although our method significantly lowers the barrier to 3D editing, it still has some limitations. First, our dataset is synthesized using the foundation model, which occasionally results in inconsistencies in certain fine details across samples. Second, since our scene completion model operates under depth constraints, it is currently unable to handle edits involving large geometric deformations. We leave these limitations as directions for future work. Nevertheless, despite these issues, we demonstrate strong performance in a wide range of scenarios, offering an effective solution for high-quality, efficient, and user-friendly 3D editing.

Clipart

Illustration
Style I

Illustration
Style II

Clay Art

Beige
Rug

Wood
Floor

Glossy
White
Furniture

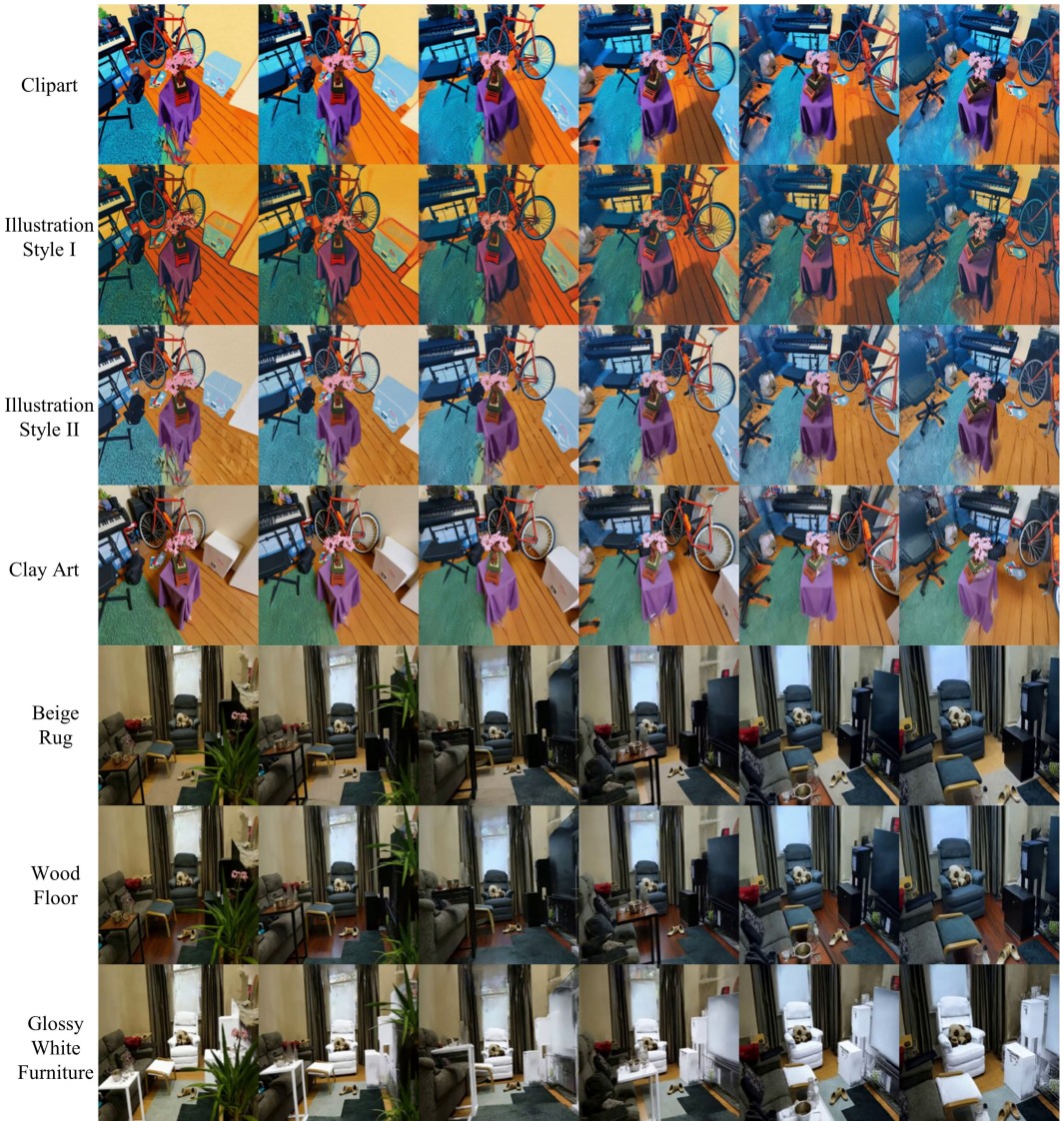

Figure S8: **Additional one-shot editing results without per-scene fine-tuning.**

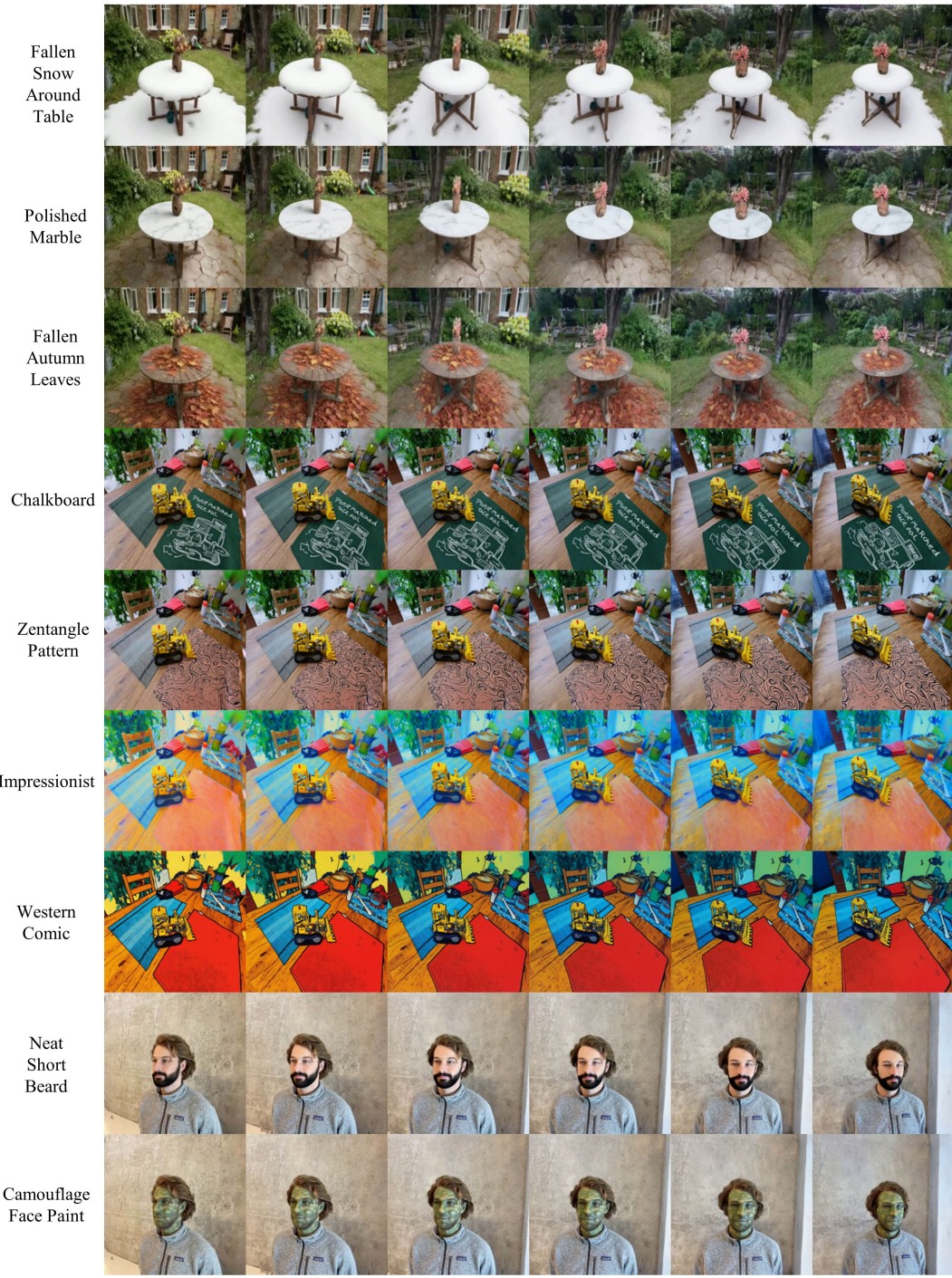

Figure S9: **Additional one-shot editing results without per-scene fine-tuning.**

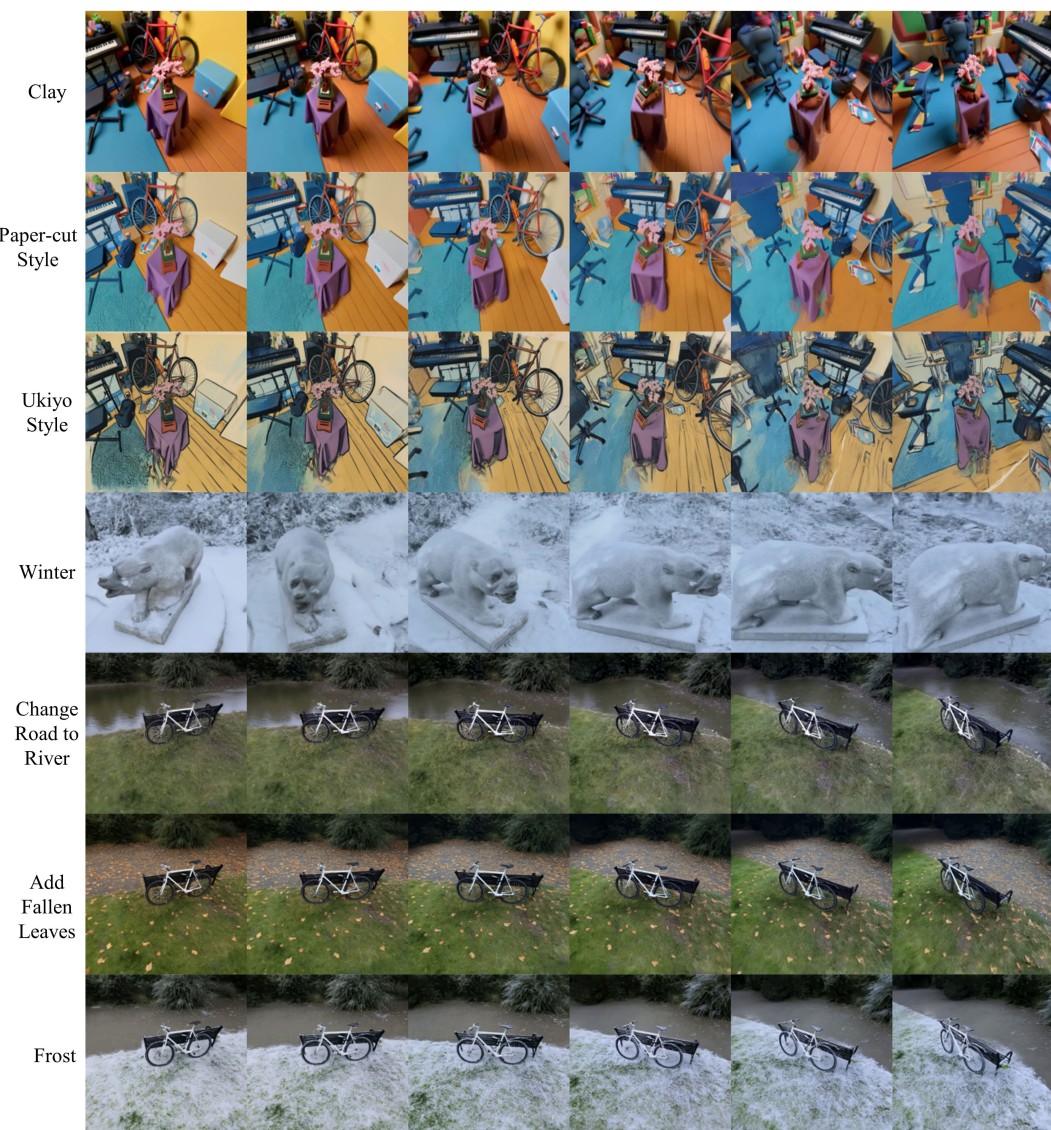

Figure S10: **Additional few-shot editing results without per-scene fine-tuning.**

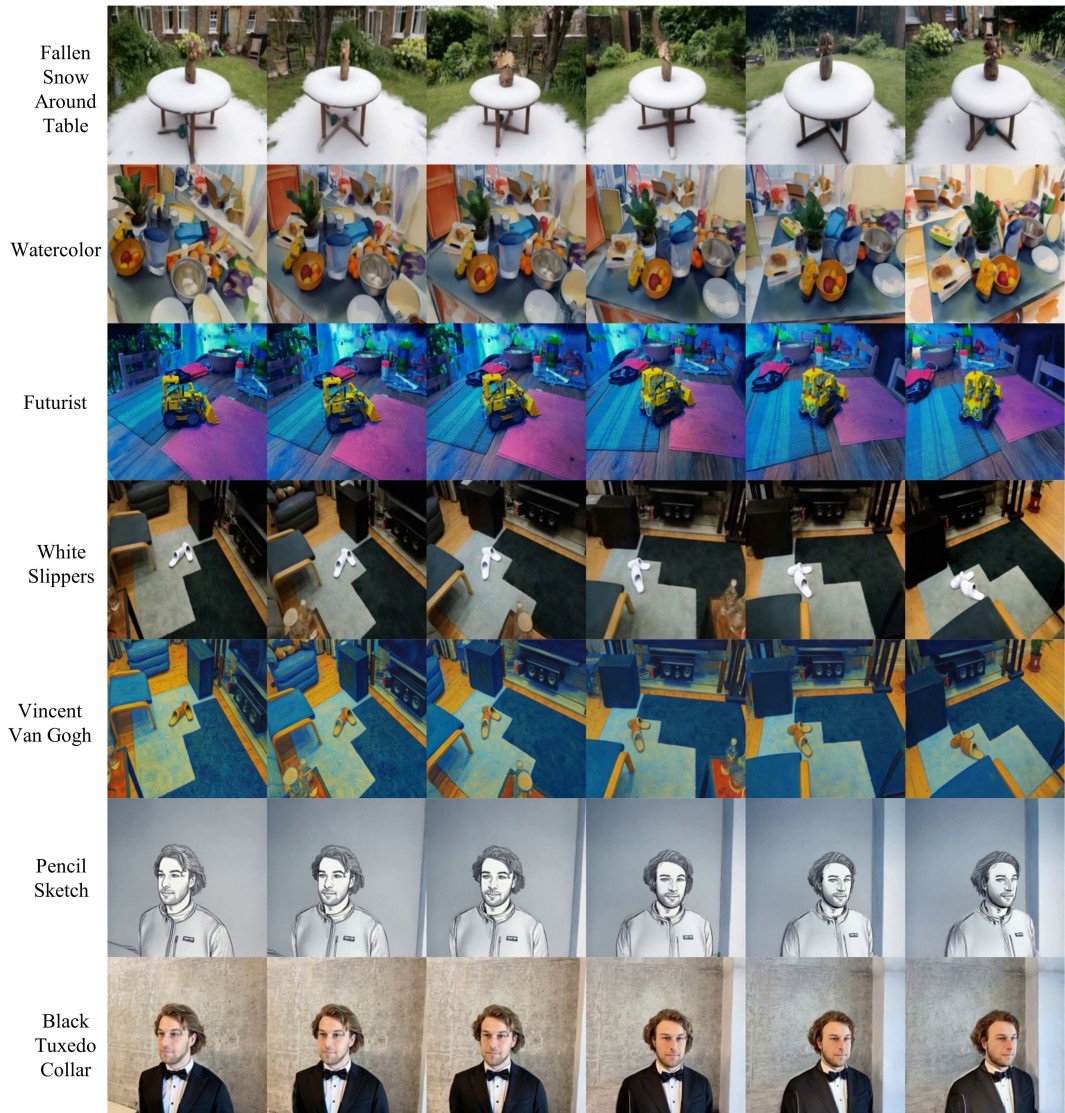

Figure S11: **Additional few-shot editing results without per-scene fine-tuning.**

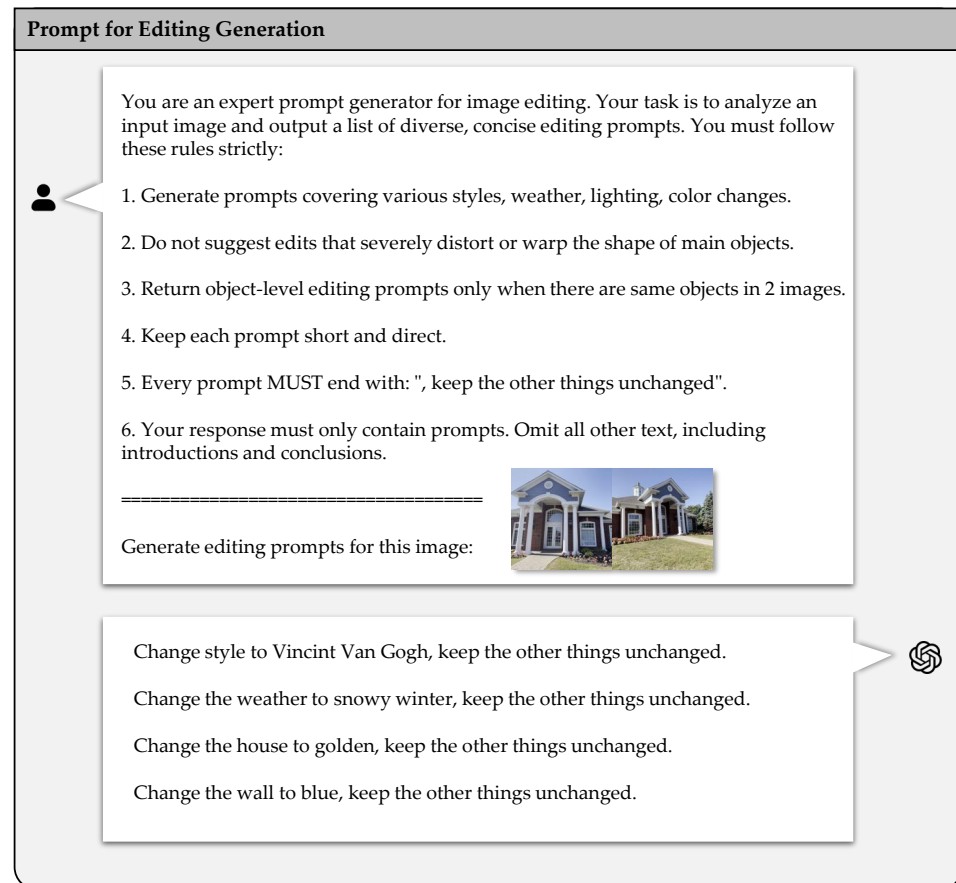

Figure S12: Input to a multi-modal large model for the generation of editing prompts.

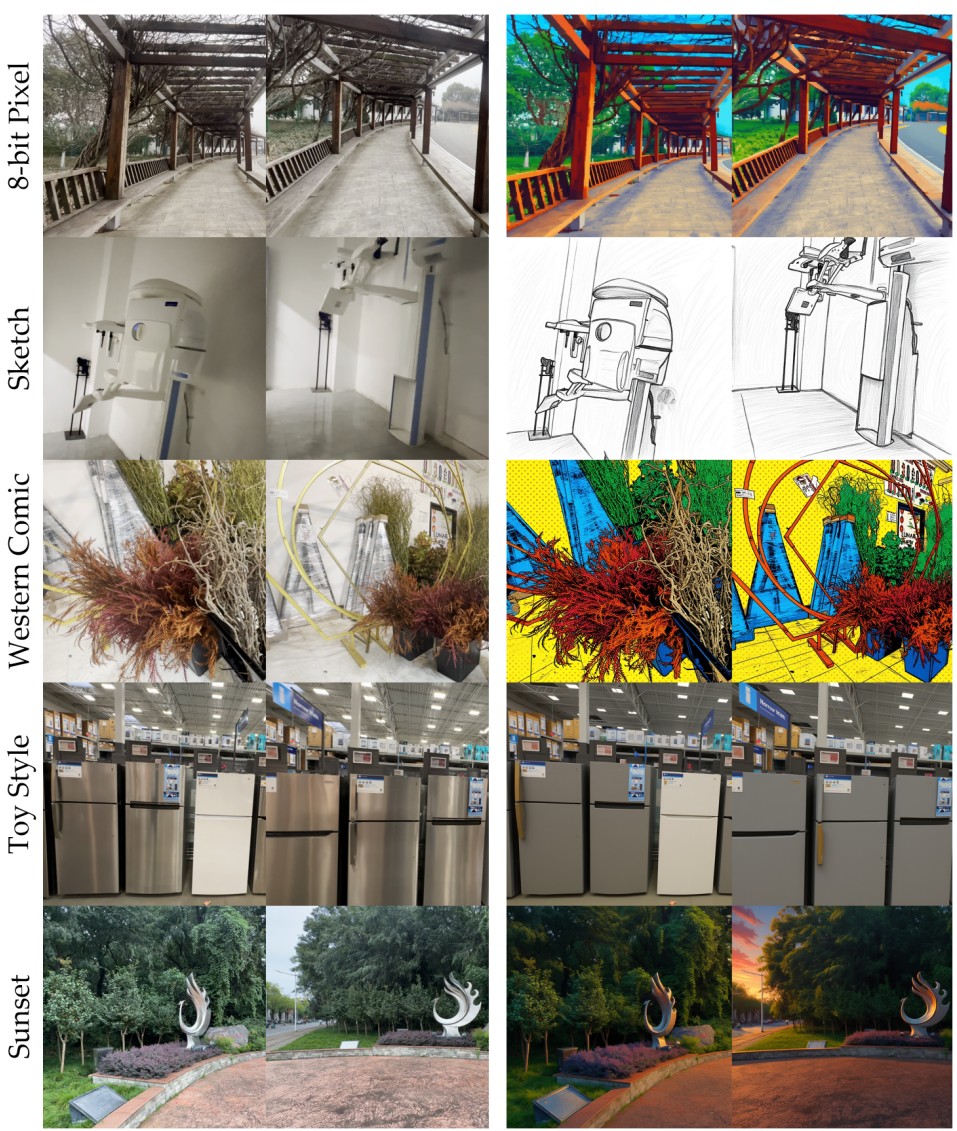

Figure S13: Examples from our synthesized multi-view consistent editing dataset. The dataset covers a wide variety of editing, including different weather conditions, lighting setups, and artistic styles.

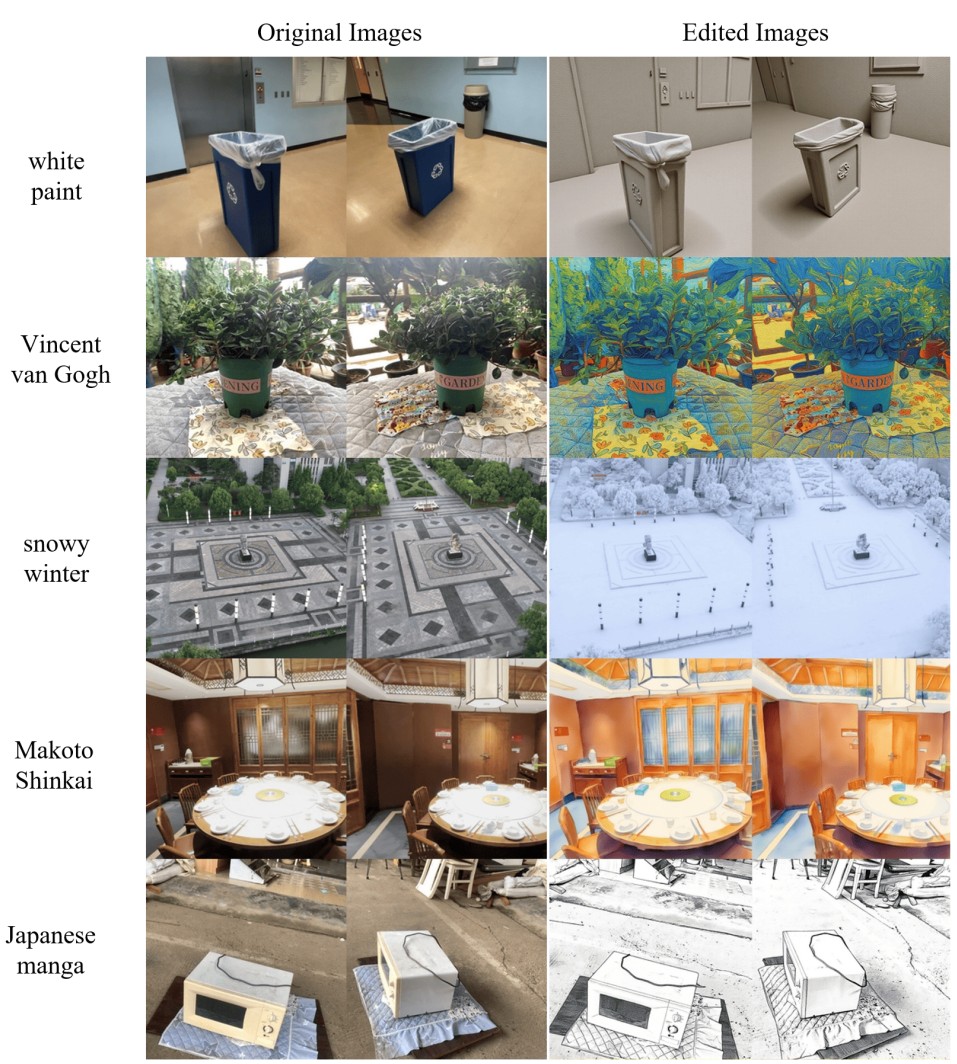

Figure S14: Additional scene-level editing data examples. Our data covers different viewpoints and styles.

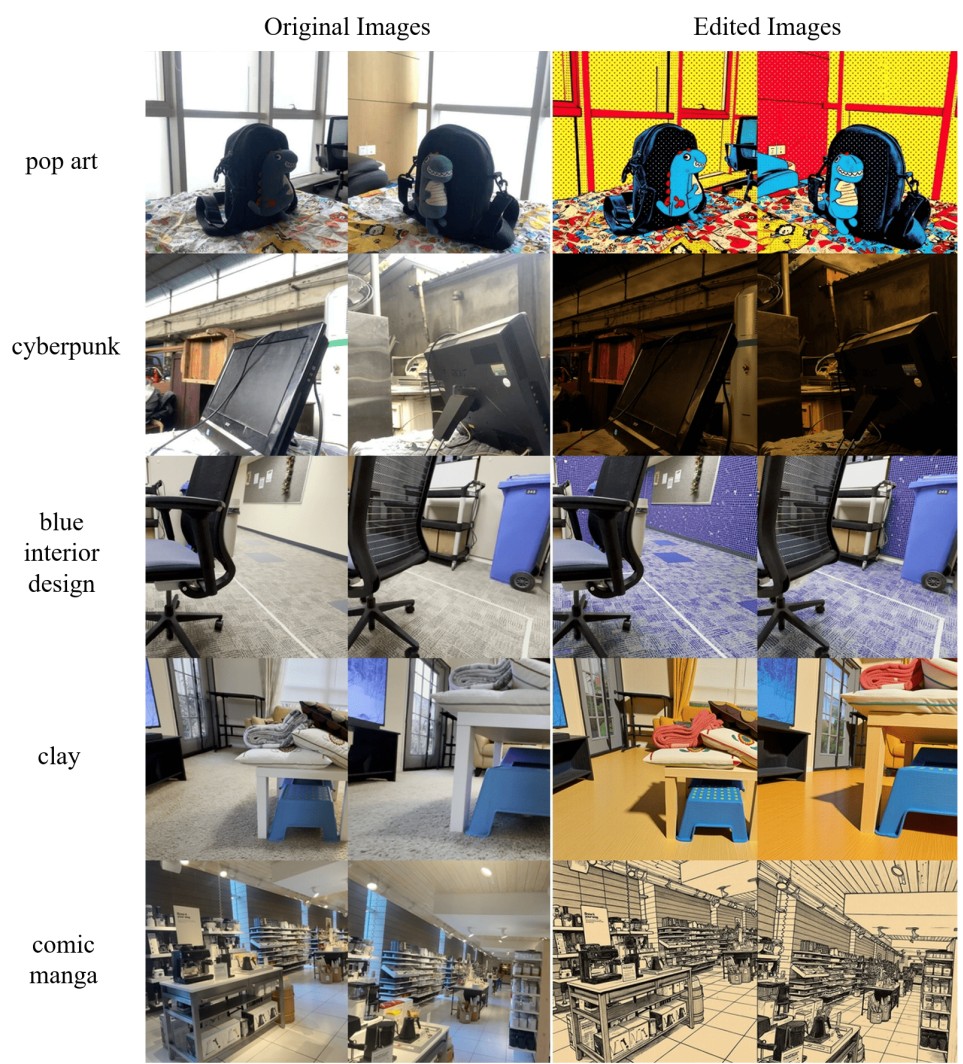

Figure S15: Additional scene-level editing data examples. Our data covers different viewpoints and styles.

Original Images          Edited Images

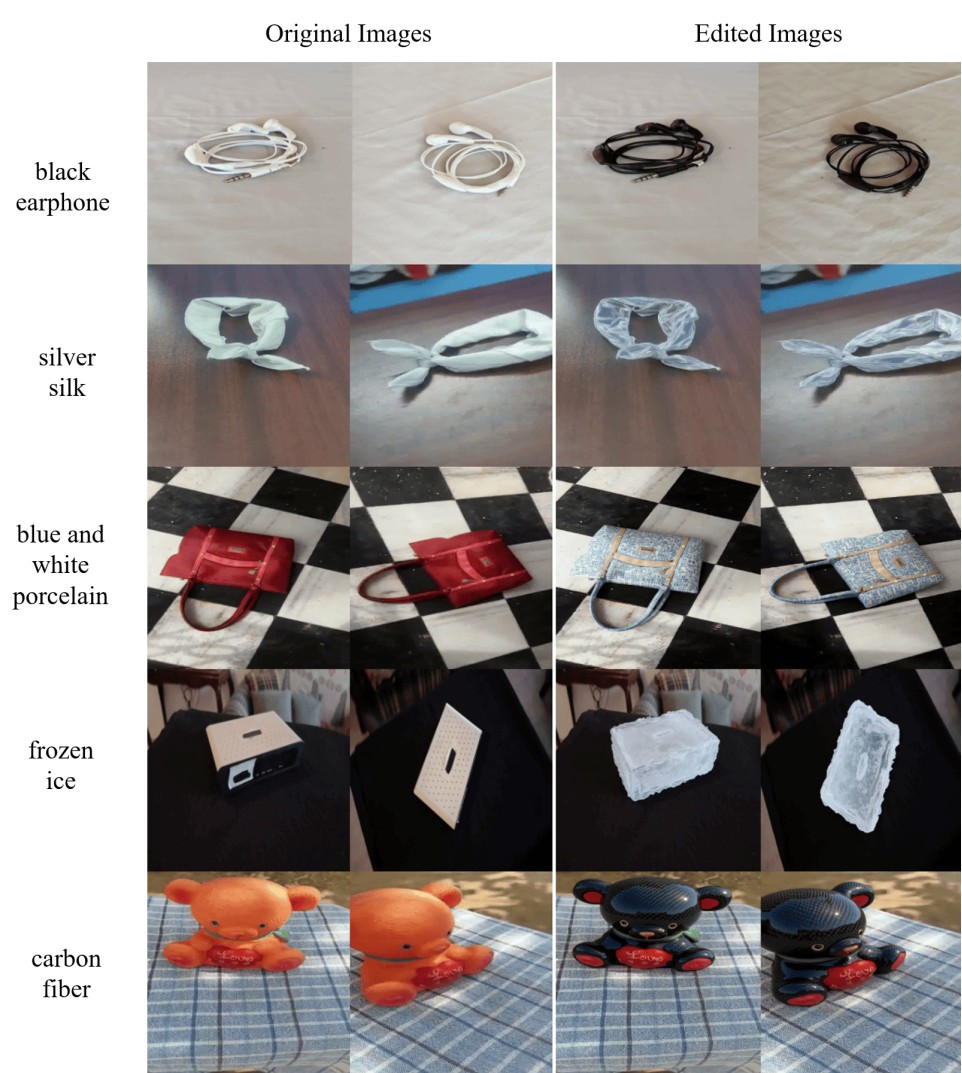

black
earphone

silver
silk

blue and
white
porcelain

frozen
ice

carbon
fiber

Figure S16:   Additional object-level editing data examples.  Our data covers different viewpoints and styles.

Original Images          Edited Images

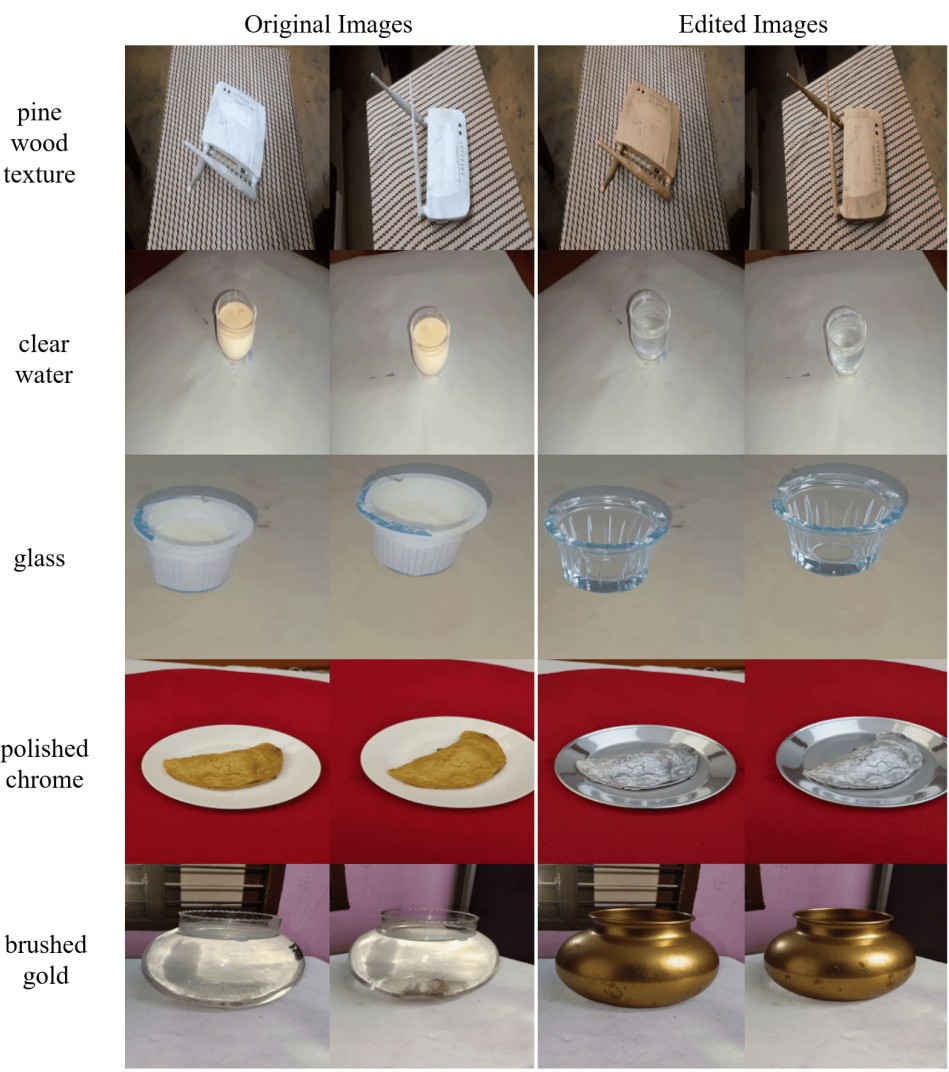

Figure S17: Additional object-level editing data examples. Our data covers different viewpoints and styles.

