# OpenReview forum: "TINKER: Diffusion's Gift to 3D--Multi-View Consistent Editing From Sparse Inputs without Per-Scene Optimization"
_ICLR.cc/2026/Conference — ICLR 2026 Poster_

### Official Review · Reviewer_RuqL · 2025-10-27

**Soundness:** 3
**Presentation:** 3
**Contribution:** 3
**Rating:** 8
**Confidence:** 4

**Summary:**

This paper proposes a feedforward 3D scene editing pipeline and introduces a multi-view consistent image editing dataset. The method adopts a state-of-the-art large image editing model to generate sparsely edited views, and then employs a finetuned video model to reconstruct the scene based on the estimated depth maps of the original scene and the sparse edited views. The results demonstrate strong multi-view consistency, and the approach does not require per-scene optimization.

**Strengths:**

1. The paper proposes a feedforward pipeline for 3D scene editing, which differs from previous per-scene optimization-based editing methods.
2. By leveraging the power of 2D editing models, Tinker preserves the identity and consistency of multi-view sparse images; moreover, by exploiting the capability of a video diffusion model, it achieves precise reconstruction by concatenating depth maps and sparse views with full attention.
3. The writing is clear and easy to follow.

**Weaknesses:**

The method appears to struggle with structural editing of scenes. In particular, it is difficult to perform large geometric changes or significant deformations. This limitation arises because, during the scene completion stage, the approach relies on the depth maps of the original videos. Such dependency introduces inconsistencies between the edited views and the original depth maps when large deformations occur, which likely leads to degraded reconstruction quality.

**Questions:**

Based on the hypothesis mentioned in the weakness section, could the authors provide further insights or explanations regarding this limitation? Specifically, can the proposed model support large geometric deformations? If not, what are the potential directions to improve or extend the framework to handle such cases? I would be happy to hear the authors’ thoughts on this.

---

> ### Author Response · Authors · 2025-11-16
>
> We thank the reviewer for the perceptive and professional feedback! We candidly acknowledge that our current framework struggles with large geometric deformations. However, we do believe this is a solvable challenge, rather than a fundamental flaw in the overall TINKER pipeline.
>
> For example, the highly inspiring recent work VIST3A [1] is very relevant. It successfully unifies a 3D reconstruction model with a 2D video generator, demonstrating the ability to directly generate high-quality 3D representations in a feed-forward manner.
>
> This direction holds immense potential.
> We believe editing with substantial geometric changes can be realized through the direct optimization of the 3D representation.
> We have already begun exploring how to integrate such a **"direct optimization on latent 3D representations"** mechanism into our TINKER pipeline. The goal would be to enhance our current scene completion model, giving it the capability to directly optimize a new 3D representation consistent with the editing instructions, rather than just completing views based on explicit depth condition.
> We believe the ultimate state of this task is that the model can achieve flexible and fine-grained controllable 3D generation and editing, just like nano-banana in the 2D image domain.
>
> We thank the reviewer again for this insightful and forward-looking question. We are very happy and grateful for the opportunity to engage in such a meaningful discussion. Please feel free to let us know if you have any further questions!
>
> [1] VIST3A: Text-to-3D by Stitching a Multi-view Reconstruction Network to a Video Generator

---

> > ### Comment · Reviewer_RuqL · 2025-11-28
> >
> > Thank you for the rebuttal and clarifications. The response aligns with my understanding of the work, and I will keep my original score.

---

> > > ### Author Response · Authors · 2025-11-28
> > >
> > > We sincerely thank you again for your time and this valuable discussion. We are grateful for this valuable opportunity to have a constructive dialogue with you!

---

### Official Review · Reviewer_yNST · 2025-10-30

**Soundness:** 2
**Presentation:** 3
**Contribution:** 2
**Rating:** 4
**Confidence:** 4

**Summary:**

The paper introduces TINKER, a framework for 3D editing that eliminates the need for per-scene diffusion model optimization, making it highly efficient compared to traditional methods. TINKER achieves multi-view consistency from as few as one or two edited images, enabling precise edits across different viewpoints.

**Strengths:**

1. The visual results are impressive and demonstrate the effectiveness of the proposed method.
2. The use of a video model for 3D editing is an innovative approach. With video generative priors, TINKER is the first method capable of jointly editing both 3D and 4D scenes.

**Weaknesses:**

1. There is an over-claim of contributions. Many baselines, such as DGE and GaussCtrl, also do not require fine-tuning the diffusion model. Therefore, this should not be considered a unique contribution of TINKER.
2. The majority of the baselines use InstructNerf2Nerf or ControlNet as the base 2D editors, whereas TINKER utilizes the FLUX model. It is unclear where the true improvement lies: is it in the advanced 2D editing model, or is it in the proposed pipeline? What if these baselines were equipped with the FLUX model?
3. AIGC tasks, such as 3D editing, necessitate a comprehensive user study to assess its performance in terms of human preference and subjective quality.
4. More visualizations of the collected training dataset should be presented to give readers a clearer understanding of the data and its characteristics.

**Questions:**

As with weaknesses

---

> ### Author Response · Authors · 2025-11-16
> **Response to Reviewer yNST (Part 1)**
>
> We sincerely thank the reviewer for the constructive and professional suggestions and feedback.
>
> **1. About the Meaning of Fine-tuning**
>
> We thank the reviewer and apologize for the ambiguity in our wording.
> **We use the term "per-scene fine-tuning" to denote not only model training but also the laborious, per-scene hyperparameter optimization required by those methods to achieve usable results.**
>
> For instance, to achieve the results shown in Figure 6, baselines like DGE and GaussCtrl require users to manually and intricately tune a large set of scene-specific hyperparameters (e.g., guidance scales, mask thresholds, number of reference views, attention blend weights ...). This is due to their underlying mechanics: DGE's base editing model relies on feature fusion guided by epipolar constraints, which makes the editing process unstable. Similarly, GaussCtrl relies on inversion sampling, making it difficult to balance editing flexibility with original scene information; consequently, it depends on per-scene optimization to obtain usable results. We found this costly, scene-by-scene tuning is necessary to strike a delicate balance between editing fidelity and structural consistency. This optimization process becomes particularly unstable and time-consuming for complex stylistic edits (e.g., "Japanese Manga" or "Vincent Van Gogh"), which often fail without such precise, per-scene adjustments.
>
> **Furthermore, we did not claim that "no diffusion model training" was our sole contribution.** As stated in our paper, TINKER's core contribution is to leverage the breakthroughs of modern DiT-based foundation models to create a generalizable framework that eliminates both forms of per-scene optimization. TINKER provides a more stable, more diverse, higher-quality, and generalizable 3D editing solution that achieves state-of-the-art performance. Most importantly, our pipeline is the first to demonstrate how these new models can be leveraged to **unify 2D editing, 3D scene editing, and even 4D editing within a single, generalizable framework.** This is what prior methods are unable to achieve.
>
> Given this, we respectfully believe our contributions are not over-claimed and represent a significant step towards scalable 3D/4D content creation. We have explicitly clarified this distinction between model training and hyperparameter optimization in our revised manuscript. We thank you again for your professional suggestions!
>
> **2. Flux on baseline methods**
>
> We thank the reviewer for this insightful question. We strongly argue that the improvement is attributable to our pipeline. Simply replacing the editor in prior works with FLUX is unviable.
>
> **2.1 Our Contribution**
>
> First, we clarify that the standalone FLUX model cannot achieve TINKER's results. FLUX-Kontext is a 2D editor; our primary contribution is the TINKER framework, which imbues a 2D model with multi-view consistent, reference-based editing capabilities. This mechanism, which is essential for 3D editing, is absent in FLUX and is a core component of our method.
>
> **2.2 Adapting Baselines with FLUX Fails**
>
> Nevertheless, to fully address the reviewer's concerns, we invested significant effort in implementing FLUX-based versions of key baselines. We call these methods:
> - Instruct-GS2GS-FLUX: An iterative dataset optimization approach similar to Instruct-NeRF2NeRF.
> - DGE-FLUX & GaussCtrl-FLUX: Methods involving multi-view feature alignment, such as Feature Injection or Feature Alignment.
> (Note: For operations requiring DDIM Inversion, we substituted the compatible RF-Solver to accommodate the flow-based architecture.)
>
> Our experiments showed that simply replacing the editor with FLUX did not lead to significant performance improvements. Instead, it introduced critical failures and prohibitive costs.
>
> **Prohibitive Computational Cost:** For iterative methods like Instruct-GS2GS-FLUX, the computational overhead became impractical. The high intrinsic inference cost of the large DiT model, when compounded by the iterative dataset refinement required by the Instruct-NeRF2NeRF paradigm, results in an extreme computational burden.
>
> **Problem of Critical Architectural Mismatch**: A more fundamental failure occurred in methods like DGE-FLUX and GaussCtrl-FLUX. These methods rely on multi-view feature alignment (e.g., DGE's STAttn, GaussCtrl's attention alignment) , which breaks down due to architectural incompatibilities between U-Net and DiT, **specifically, the positional embedding (PE).** In U-Net: Attention features generally do not contain explicit positional encoding. This allows cross-view attention to match features based purely on content similarity. In DiT: Positional encoding is deeply integrated into the attention blocks. When naively adapting the baselines, the introduction of the DiT's strong PE fundamentally disrupts the model's ability to understand semantic content during multi-view alignment with prior attention designs.

---

> ### Author Response · Authors · 2025-11-16
> **Response to Reviewer yNST (Part 2)**
>
> Consequently, Our comprehensive experiments confirm that the observed improvements are not merely an "editor upgrade." The attention designs of prior methods are architecturally coupled to the U-Net design, and their core mechanisms cannot be trivially upgraded by substituting the 2D editor. Conversely, TINKER demonstrates superior quality and flexibility, supporting 2D, 3D, and 4D editing, which traditional methods are unable to achieve.
>
> **3. User Study**
>
> We thank the reviewer for this valuable suggestion. Following this suggestion, we have conducted a comprehensive user study to evaluate the perceptual quality of TINKER against the baseline methods.
> We recruited 50 participants for the study. They were asked to evaluate the results from 20 different editing scenes (covering both object-level and scene-level edits) generated by our method and the baselines. Participants were asked to rate each result on a 5-point Likert scale (1 = Very Poor, 5 = Very Good) across three key dimensions:
>
> - Text Similarity: How well the edited 3D scene matches the text prompt.
> - Editing Quality: The visual fidelity, realism, and aesthetic quality of the edit, and the absence of visual artifacts.
> - Multi-View Consistency: How consistent and stable the edit appears when rendered from various novel viewpoints.
>
> As the results in the table below clearly indicate, TINKER significantly outperforms all baseline methods in all three subjective categories. This strong preference from human evaluators provides compelling evidence that TINKER not only achieves superior quantitative metrics but also produces 3D editing results that are demonstrably more faithful to the user's intent.
>
> | | Text similarity | Quality | Consistency |
> |---|---|---|---|
> | GaussCtrl | 3.62 | 3.79 | 3.87 |
> | DGE | 3.78 | 3.44 | 3.73 |
> | EditSplat | 3.17 | 2.92 | 3.42 |
> | TIP-Editor | 2.57 | 2.89 | 3.56 |
> | Ours-Oneshot | 4.38 | 4.49 | 4.31 |
> | Ours-Fewshot | **4.52** | **4.61** | **4.55** |
>
> **4. More Data Visualizations**
>
> Thank you for this valuable suggestion.In the original manuscript, we provided a range of visualization samples in Figure S12 (S13 in the revised version). To more comprehensively provide readers with a clearer understanding of our data, we have added 4 new figures (Figure S14, S15, S16, S17) to the appendix of our revised paper. These figures include more convincing dataset visualizations of both object-level and scene-level edits. We believe these additional visualizations further validate the effectiveness of our data generation pipeline and compellingly demonstrate our dataset's diversity and quality.
>
> We thank the reviewer again for the constructive and professional feedback! All the revisions are marked red in the revised paper. Please let us know if you have any further concerns. Any further discussion is warmly welcomed!

---

> > ### Comment · Reviewer_yNST · 2025-11-27
> >
> > Thanks for the detailed clarification provided by the author. All my concerns are resolved, and I have decided to increase my score. Please remember to include those detailed discussions in the final paper to improve the soundness.

---

> > > ### Author Response · Authors · 2025-11-28
> > >
> > > We certainly will! We would like to express our gratitude once again for your professional feedback and comments. We truly appreciate this valuable opportunity to engage in a discussion with you!

---

### Official Review · Reviewer_KqtQ · 2025-10-30

**Soundness:** 2
**Presentation:** 3
**Contribution:** 2
**Rating:** 6
**Confidence:** 4

**Summary:**

This paper introduces a framework for 3D editing that operates without per-scene fine-tuning. The approach consists of two core components: a multi-view consistent editor, built upon a large-scale image editing model and a multi-view image editing dataset, and a scene completion model that generates dense edited views. The method is evaluated on the Mip-NeRF-360 and IN2N datasets with qualitative and quantitative results.

**Strengths:**

1. This paper introduces a one-shot or few-shot approach for 3D editing.
2. This paper proposes a generalizable pipeline for 3D editing.
3. The paper is well-structured and easy to follow.

**Weaknesses:**

1. For the multi-view image editing model, when dealing with views that have large variations, does the multi-view consistency of the edits decrease? If so, could these inconsistencies be further propagated and amplified by the subsequent scene completion model?
2. Since the scene completion model relies on geometric information like depth maps, in few-shot or even one-shot settings with large view variations, is it prone to introducing more hallucinations or geometric distortions to fill in the missing information?
3. Although the method eliminates per-scene finetuning, its overall editing time, as shown in Table 1, does not present a significant advantage over some baseline methods. This suggests that the inference cost of the models involved might be a bottleneck.
4. In the supplementary video "Edited_Novel_View_Rendering.mp4" for the IN2N person scene, noticeable artifacts can be observed around the edges of the edited object. What is the primary cause of these edge artifacts?

**Questions:**

Please refer to the weaknesses.

---

> ### Author Response · Authors · 2025-11-16
> **Response to Reviewer KqtQ (Part 1)**
>
> We sincerely thank you for your professional questions and constructive feedback!
>
> **1. Large Variations Are Actually Beneficial**
>
> We thank the reviewer for this insightful and professional question. When handling views with large variations, the multi-view consistency of our edits does not decrease. **In fact, these high-quality, large-variation views actively help our scene completion model achieve better results.**
>
> **Robustness of the Editing Model:** Firstly, our multi-view consistent editing model is capable of generating high-quality and geometrically consistent results even across large viewpoint changes. This robustness not only stems from the strong prior of the base editing model, but also from our specific fine-tuning dataset, which was deliberately constructed using multi-view data. This dataset includes many training samples with significant viewpoint differences, which explicitly trained the model to maintain consistency in this challenging setting.
>
> **Large Variation's Benefit to the Scene Completion Model:** More Significantly, these consistent edits from widely different viewpoints actually provide more valuable and complementary spatial information to our scene completion model than edits from very similar views. When sparse reference views are too close, their information is highly redundant and offers limited new cues for the rest of the scene. In contrast, reference views with large separation introduce more information gain, providing diverse geometric and textural details from new perspectives. This richer, less-redundant information allows the scene completion model to reconstruct the full scene's structure with higher fidelity and consistency.
>
> **New Experimental Validation:** We have conducted a new ablation study to prove this point, which will be added to the appendix. For the same original video, we provided our scene completion model with only two edited reference views as input. We then varied the frame interval (and thus the angular separation) between these two views. As shown in the table below, the final reconstruction quality and consistency of the completed scene improve as the viewpoint separation between the two reference frames increases. This confirms that large-variation views are a benefit, not a liability, for our pipeline.
>
> | | Text-dir | DINO | Aesthetic |
> |---|---|---|---|
> | 10 degree - 2 views | 0.143 | 0.955 | 6.216 |
> | 45 degree - 2 views | 0.148 | 0.956 | 6.335 |
> | 90 degree - 2 views | 0.155 | 0.959 | 6.427 |
>
> **2. Depth MapsHelp to Reduce Hallucinations**
>
> We thank the reviewer for this valuable question. In TINKER, the depth map does not introduce hallucinations; on the contrary, it is the most critical constraint we use to prevent hallucinations and geometric distortion.
> In any sparse-view scene completion task, the fundamental challenge is ambiguity. When a model must fill large, unobserved regions with no structural guidance, it is forced to guess the content (e.g., whether a region should be flat ground, an object, or the sky). This guesswork is the primary source of hallucinations.
>
> Our scene completion model is explicitly designed to solve this ambiguity by learning to reason based on geometry. We achieve this through extensive depth-conditioned training on large-scale 3D-centric datasets. This is crucial: the model learns a powerful prior of geometric-semantic knowledge. It is no longer blindly in-painting pixels; it is reasoning about the 3D structure provided by the depth map. For example, the model learns that a specific depth profile corresponds to a tree and generates tree-like textures, rather than hallucinating a building.
>
> This is precisely what we demonstrated in our ablation study in Figure S3 and Table S1:
> - Methods that lack strong geometric constraints, such as Framer or the Ray Map-conditioned baseline, must guess the content for missing views. As our results show, this leads to catastrophic content hallucinations and structural distortions.
> - In stark contrast, our depth-conditioned method remains geometrically faithful and consistent. The depth map acts as an explicit structural prior, guiding the model to generate a reasonable and coherent completion that honors the underlying geometry of the scene, even from sparse inputs.
>
> Therefore, in our framework, the depth map is the key solution that ensures geometric fidelity in few-shot and one-shot settings, rather than a source of distortion.

---

> > ### Author Response · Authors · 2025-11-16
> > **Response to Reviewer KqtQ (Part 2)**
> >
> > **3. For Inference Cost**
> >
> > We thank the reviewer for this valuable feedback. We agree that the current inference cost is a bottleneck and a key area for future improvement. However, we would like to offer two important points of context.
> >
> > **Performance vs. Cost:** While our inference time is comparable to some baselines, those methods rely on significantly smaller U-Net-based architectures. The fact that their complex pipelines still require similar time, despite using less powerful models, highlights the inherent inefficiencies and limitations TINKER was designed to solve. TINKER leverages a much larger model to achieve significantly superior quality and consistency. The current runtime is a trade-off for this substantial leap in quality and generalizability.
> >
> > **Scalability and Benefit from New Acceleration Techniques:** More importantly, our choice of modern DiT architecture is a strategic advantage. Unlike the mature U-Net paradigm, this architecture is the focus of intense, ongoing community optimization and benefits from a rapidly evolving ecosystem of acceleration techniques. This includes General-purpose optimizations that dramatically reduce memory and computational overhead, such as FlashAttention-2[1] and SageAttention2[2], and diffusion-specific acceleration methods designed explicitly for these new models. For example, the recent FPSAttention [3] (NeurIPS 2025 Spotlight) demonstrated a 2.45x inference speedup on a 1.3B video diffusion model, an architecture we also used as our scene completion backbone (WAN2.1 1.3B), with no quality degradation.
> >
> > Therefore, we believe the inference cost shown in Table 1 is not a limitation that can't be addressed. TINKER is designed to directly benefit from these and future optimizations, allowing its inference cost to be substantially reduced. This architectural choice highlights the scalability and forward-looking nature of our framework, which will only become more efficient as the underlying models and inference engines improve.
> >
> > [1] FlashAttention-2: Faster Attention with Better Parallelism and Work Partitioning
> > [2] SageAttention2: Efficient Attention with Thorough Outlier Smoothing and Per-thread INT4 Quantization
> > [3] FPSAttention: Training-Aware FP8 and Sparsity Co-Design for Fast Video Diffusion
> >
> >
> > **4. Small Edge Artifacts**
> >
> > By default, we initialize the edited scene using the Gaussians from the original scene. Furthermore, we use a single, fixed set of 3DGS optimization hyperparameters for all examples presented, regardless of the editing type or intensity. Consequently, in a few cases, residual information from the original scene may be left, especially for boundaries. These default parameters are not perfectly optimal for the given edit, leading to the minor edge artifacts. This can be fully resolved by simply increasing the number of 3DGS training iterations. We have updated the results in the Supplementary Materials.
> >
> >
> > We sincerely thank you once more for your invaluable questions and constructive feedback! We are very grateful for this valuable opportunity to discuss our work with you. Please let us know if you have any further concerns and suggestions.

---

### Official Review · Reviewer_c8ha · 2025-10-30

**Soundness:** 4
**Presentation:** 4
**Contribution:** 4
**Rating:** 6
**Confidence:** 3

**Summary:**

This paper, Tinker, has proosed a general-purpose 3D editing framework. Given the reconstructed 3DGS, the proposed editing pipelien will perform video depth estimation, muti-view consistent editsing, and scene completion sequentially. The edited 3dgs is of high quality under comprehensive evaluation.

**Strengths:**

1. The proposed method is efficient, reasonable, and offers high quality.
2. The writing is good.
3. The proposed dataset will be very helpful to this field.

**Weaknesses:**

1. The main issue is that this method looks very complicated, though it is necessary to make the 3D editing feed-forward. Still, too many components are involved in this process.

Overall, I think this is a good paper and worth acceptance. I just encourage the authors to think of the next step and tackle this task in a more elegant way.

**Questions:**

1. Since VDM has made great progress recently, 3D-aware VDM like Lyra, Gen3C, and VIST3A has also shown good capability. I wonder whether the proposed pipeline can be radically replaced by the VDM-based pipeline.
2. Besides, 3D foundation models are getting better now. Rather than directly working on the 3DGS, I wonder whether the proposed pipeline can be improved to incorporate 3D VFMs like VGG-T / AnySplat to facilitate easier 3D reconstruction editing.

---

> ### Author Response · Authors · 2025-11-16
>
> We sincerely thank you for your insightful and professional feedback!
>
> **W1. The method looks complicated.**
>
> We agree that the pipeline involves multiple components. While our method may appear complex at first glance, the core idea is actually simple: we first perform consistent image editing to obtain sparse edited views, then use a video model to densify them into a sequence, and finally optimize the Gaussian representation. We appreciate the encouragement to explore more elegant solutions, which aligns perfectly with our future works for the next steps, as discussed below:
>
> **Q1. Relations to 3D-aware VDMs**
>
> We appreciate your insightful questions and fully share your interest in the rapid progress of 3D foundation models and 3D-aware VDMs, including Lyra, Gen3C, and VIST3A. These works are also among the directions we are closely tracking.
>
> **We think that TINKER and emerging 3D-aware VDMs are not competitive but are, in fact, synergistic.** Their relationship is mutual benefit, not replacement. This distinction arises from their current objectives: TINKER is engineered as a general-purpose 3D editing framework, whereas 3D-aware VDMs are primarily focused on 3D asset generation. Given their distinct goals, they currently operate in complementary, non-overlapping domains.
> We argue they are co-evolving and mutually reinforcing, much like the development of 2D image generation and editing models. We envision the 3D domain will ultimately converge towards a flexible, unified paradigm for generation and editing, analogous to models like "nano-banana" in image generation and editing.
>
> The aforementioned 3D-aware VDMs illuminate a potential pathway toward this goal. **Inspired by this, we are now actively exploring the incorporation of feed-forward 3D editing into the TINKER paradigm.** This direction is highly promising, as it not only reduces inference costs but also offers a viable path to address our current limitations with large-scale geometric deformations. Critically, it also provides a framework that can be unified with contemporary 3D asset generation methods. Our preliminary experiments have already shown the significant potential of this unified feed-forward approach.
>
> Finally, for the task of feed-forward 3D editing, a robust pipeline to construct large-scale 3D asset editing pairs as data remains a fundamental challenge. This is just the problem that TINKER is able to solve. Consequently, we believe TINKER is an indispensable method for the future of unified feed-forward 3D generation and editing.
>
> To sum up, while 3D-aware VDMs offer crucial insights of high-quality feed-forward 3D asset generation to inspire TINKER's evolution towards feed-forward editing, TINKER complements their inability to edit and provides a effective data-construction pipeline required for a future, unified feed-forward 3D generation and editing paradigm.
>
> **Q2. Incorporating 3D foundation models**
>
> We agree that 3D foundation models will be key enablers for future Unified Feed Forward Generation and Editing pipeline. Recent work such as Geometry Forcing [1] demonstrates that integrating VGGT features into video diffusion can significantly improve geometric consistency. We believe that leveraging strong 3D foundation model features will substantially enhance the quality of feed-forward pipelines. Moreover, we believe different 3D decoders can be designed to support representations beyond 3DGS, such as point clouds or meshes, making the system more suitable for practical applications in graphics (e.g., Blender) and robotics, where high-quality 3D assets are in high demand.
> Furthermore, we view generation and understanding as inseparable tasks. We believe these two tasks will be both compatible and mutually reinforcing.
>
> [1] Geometry Forcing: Marrying Video Diffusion and 3D Representation for Consistent World Modeling.
>
>
> We sincerely thank you again for your invaluable questions and we sincerely appreciate this opportunity to discuss our work with you! Please let us know if you have any further concerns and suggestions.

---

### Author Response · Authors · 2025-11-16

Dear reviewers, ACs, SACs, PCs,

We sincerely thank all reviewers for their dedicated time and insightful feedback on our manuscript. We are grateful for the constructive comments and the positive recognition of our work.

We are highly encouraged to note that the reviewers share our focus on the same key future directions, such as Feed-Forward 3D Gaussian. We are grateful for this insightful engagement and the opportunity to discuss the future evolution and potential improvements for TINKER with **Reviewer c8ha** and **Reviewer RuqL**.

We are particularly grateful to **Reviewer KqtQ** and **Reviewer yNST**, who provided not only highly professional critiques but also invaluable suggestions for enhancing TINKER.
**Reviewer KqtQ** raised several great points, including: (1) the impact of view variations, (2) the potential for depth map-induced hallucinations, (3) concerns over inference cost, and (4) an inquiry into a specific edge artifact. We have thoroughly addressed each of these in our individual response, providing detailed explanations and new supporting experiments. In our response, we demonstrated that large view variations, rather than being detrimental, actually provide greater informational gain that enhances model performance. We also clarified that the use of depth maps is a key factor in mitigating model-induced hallucinations. Regarding practical concerns, we explained that TINKER's inference cost can be readily optimized by leveraging existing acceleration techniques. Finally, we clarified that the noted edge artifact is not a fundamental limitation of our method and can be straightforwardly resolved.

**Reviewer yNST** identified an imprecision in our "per-scene fine-tune" terminology. We have corrected this in the revised paper, explicitly stating that this process encompasses both model training and costly hyper-parameter tuning. Furthermore, in response to the suggestions, we have incorporated: (1) comprehensive experiments about flux-adapted baselines, (2) a new User Study to validate that TINKER's results are strongly preferred by users, and (3) more data visualizations to improve clarity.

The professionalism and insightful feedback from all the reviewers have been instrumental in significantly improving the quality and rigor of our work. Finally, we hope to express our sincere gratitude again to all reviewers, as well as to the ACs, SACs, and PCs, for their substantial time and dedicated effort throughout this review process.

---

### Comment · Area_Chair_jwds · 2025-11-27

Dear Reviewers,

As we enter the discussion phase, I strongly encourage you to read the authors' rebuttal carefully and acknowledge their effort. Silence is the worst outcome for an author. Even if the rebuttal does not change your final rating, a brief response explaining why the concerns remain unaddressed is crucial for a fair process. Please help us make an informed decision by engaging in a constructive dialogue.

AC

---

### Meta-Review · Area_Chair_NAgT · 2026-01-02

**Summary:**

The paper initially received one negative and three positive ratings. The concerns are mostly about 1) complexity of the proposed method, 2) sensitivity to depth estimation and multi-view variations, 3) some technical clarifications, e.g., finetuning strategy, base model.

**Reviewer Concerns:**

The authors have provided responses in the rebuttal to answer initial concerns from the reviewers. The AC took a close look at the paper, reviews, and the rebuttal. After the rebuttal, the AC finds that most questions are addressed well, especially the further clarification of technical contributions and additional experiments. This also leads to the increased rating from reviewer yNST to be positive. Therefore, the AC agrees with the reviewers' overall feedback and hence recommends the acceptance rating, while strongly encouraging the authors to revise the paper accordingly and release the code for reproducibility.

**Reviewer Scores:**

Reviewer yNST mentioned raising the original rating from 4, and reviewer RuqL indicated to retain the original positive rating, while the other two reviewers did not fully participate in the discussion.

---

### Decision · Program_Chairs · 2026-01-26

Accept (Poster)